# Sensory perception of dead conspecifics induces aversive cues and modulates lifespan through serotonin in Drosophila

Tuhin S. Chakraborty[1,3], Christi M. Gendron[1,3], Yang Lyu [1], Allyson S. Munneke[2], Madeline N. DeMarco[1], Zachary W. Hoisington[1] & Scott D. Pletcher[1,2]

Sensory perception modulates health and aging across taxa. Understanding the nature of relevant cues and the mechanisms underlying their action may lead to novel interventions that improve the length and quality of life. We found that in the vinegar fly, *Drosophila melanogaster*, exposure to dead conspecifics in the environment induced cues that were aversive to other flies, modulated physiology, and impaired longevity. The effects of exposure to dead conspecifics on aversiveness and lifespan required visual and olfactory function in the exposed flies. Furthermore, the sight of dead flies was sufficient to produce aversive cues and to induce changes in the head metabolome. Genetic and pharmacologic attenuation of serotonergic signaling eliminated the effects of exposure on aversiveness and lifespan. Our results indicate that *Drosophila* have an ability to perceive dead conspecifics in their environment and suggest conserved mechanistic links between neural state, health, and aging; the roots of which might be unearthed using invertebrate model systems.

---

[1] Department of Molecular and Integrative Physiology and Geriatrics Center, University of Michigan, Ann Arbor, MI 48109, USA. [2] Program in Cellular and Molecular Biology, University of Michigan, Ann Arbor, MI 48109, USA. [3]These authors contributed equally: Tuhin S. Chakraborty, Christi M. Gendron. Correspondence and requests for materials should be addressed to S.D.P. (email: spletch@umich.edu)

Sensory perception influences energy homeostasis, tissue physiology, and organism aging through neuronal circuits that emanate from sensory tissues and that interface with deeper regions of the central nervous system. The molecular nature of these relationships was first described in the nematode, *Caenorhabditis elegans*[1], and sensory effects on aging have been observed across the phylogeny of vertebrate and invertebrate animals[2–8]. Sensory inputs relate information about nutrient availability and reproductive opportunity to rapidly initiate physiological changes that occur in coordination with known behavioral outcomes, suggesting similarities in the underlying circuitry. Conserved neuromodulators, including biogenic amines and neuropeptides, that influence responses to food and mates are known to also modulate lifespan in a state-dependent manner[8–13].

The ability to perceive dead individuals is not exceptional in the animal kingdom, as individuals from a range of species respond to dead conspecifics with a variety of different effects. Social insects, including ants and honey bees, exhibit necrophoresis, in which dead colony members are systematically removed from the nest to promote hygienic conditions[14]. Dead zebrafish scents provoke defensive behavior in live individuals[15], and the sight of a dead conspecific induces alarm calling in scrub-jays[16], suggesting that dead individuals may indicate danger. Elephants and nonhuman primates exhibit stereotypical behaviors toward dead individuals associated with permanent loss of a group member[17,18]. In humans, the effects of experiences with death include emotional dysregulation and depression, as well as physical effects such as headaches, fatigue, and cardiovascular disease[19–21].

In some cases, the sensory mechanisms through which individuals perceive dead conspecifics are known (e.g., refs. [14,22–24]), but to our knowledge, the extent to which these experiences influence aging and, if so, the degree to which the effects are shared across species are yet to be determined. In this study, we provide evidence that exposure of *Drosophila melanogaster* to dead conspecifics (i) induces cues in the exposed flies that are aversive to other non-exposed flies, (ii) modulates several physiological parameters, including the abundance of stored lipid, respiration rate, and climbing ability, and (iii) reduces lifespan. These behavioral and physiological effects are likely mediated by sensory perception because our observed phenotypes required visual cues and were modulated by olfactory cues. Furthermore, the sight of dead flies, but not their smell or taste, was sufficient to induce the production of aversive cues. The negative effects of exposure to dead conspecifics were reversed by targeted pharmacologic and genetic attenuation of serotonin signaling, suggesting the possibility that such effects are conserved in other taxa.

## Results

### *Drosophila* exposed to dead conspecifics induce aversive cues.

While investigating whether adult *Drosophila* behaviorally responds to diseased individuals in their environment, we discovered that flies show an aversive response after exposure to dead conspecifics. In our initial experiments, we established a binary choice assay (T-maze) in which flies that were previously infected with the lethal pathogen *Pseudomonas aeruginosa PLCS* were placed behind a screen in one side of a T-maze and healthy flies were placed in the opposite side. When naive choosers were loaded into the T-maze, we found that they sorted non-randomly, in that they avoided the side of the T-maze containing a group of flies that had been infected 24–48 h previously (Supplementary Fig. 1a). We consistently failed to observe avoidance in naive choosers to groups of flies that had been infected for <24 h. Flies

began dying from our *Pseudomonas* infection roughly 24 h post infection, suggesting that the appearance of dead flies rather than infection might be the cause for the aversion. We therefore asked whether dead flies alone were sufficient to create an aversive stimulus. We found that they were not (Supplementary Fig. 1b). When comparing preference between only healthy live flies vs. only dead flies, naive choosers preferred dead flies (Supplementary Fig. 1c), presumably due to $CO_2$ emitted from live animals, which is a known repulsive stimulus[25], establishing that the dead animals themselves were not intrinsically aversive. We therefore asked whether a mixture of dead flies with healthy live flies was aversive compared to healthy live flies alone. We observed a strong preference of naive flies choosing the side of the T-maze without dead animals (Supplementary Fig. 1d). Finally, healthy flies from different laboratory strains that had been pre-exposed to dead conspecifics for 48 h (the dead flies were removed immediately prior to the choice assay) exhibited aversive qualities (Fig. 1a, b), establishing that the presence of dead flies in the T-maze was not required for aversion. Together, these data indicated that dead fly exposure triggered changes in healthy live individuals that repelled naive choosers.

Control experiments ruled out positional artifacts and biases in our technical apparatus as causes for the preferred segregation of naive choosers away from flies exposed to dead animals. Naive choosing flies segregated randomly when both sides of the T-maze were empty or when both sides contained equal numbers of live, unexposed flies (Supplementary Fig. 2a and b). Aversive cues were not induced when flies were mock-exposed to dead animals using black beads that are roughly the size of a fly, and the presence of dead flies did not affect feeding over 24 h (Supplementary Fig. 2c and d). This suggests that aversive cues in exposed flies are not triggered by structured environments or by changes in food accessibility. We also tested whether exposed flies were exhibiting a response to perceived increases in population density by augmenting the number of live flies in the unexposed treatment so that the total number of flies in the pre-choice environments was equal. We observed no induction of aversive cues, thus ruling out density effects (Supplementary Fig. 2e).

### Factors that affect the aversive response.

Subsequent experiments using only healthy animals pre-exposed to dead individuals before behavioral testing revealed that the effects of dead fly exposure are reproducible and are influenced by the characteristics and/or reason for death of the of the dead flies. Exposed flies lost their aversive characteristics approximately 10 min after dead flies were removed (Fig. 1c), which is approximately the time span of short-term memory in *Drosophila*[26], indicating that the aversive effect is persistent but short-lived. The aversiveness of exposed flies was affected by the number of dead flies included in the environment and the duration of exposure: exposed flies became more aversive as the time of exposure and the number of dead in the environment increased, although in standard rearing vials the magnitude of the effect saturated at roughly 48 h and 10 dead animals, respectively (Fig. 1d, e). Flies exposed for 48 h to flies that died from starvation or from normal aging triggered aversive cues, while a similar exposure to flies killed by immersion in liquid nitrogen did not (Fig. 1f). Flies that had died 46 days prior to testing also failed to induce aversive cues in exposed animals (Fig. 1g).

We tested whether the effects of exposure to dead animals would be influenced by the evolutionary relatedness between the dead and live animals by exposing *Drosophila melanogaster* to dead individuals from one of three related species (*Drosophila virilis*, *Drosophila simulans*, and *Drosophila erecta*). We found

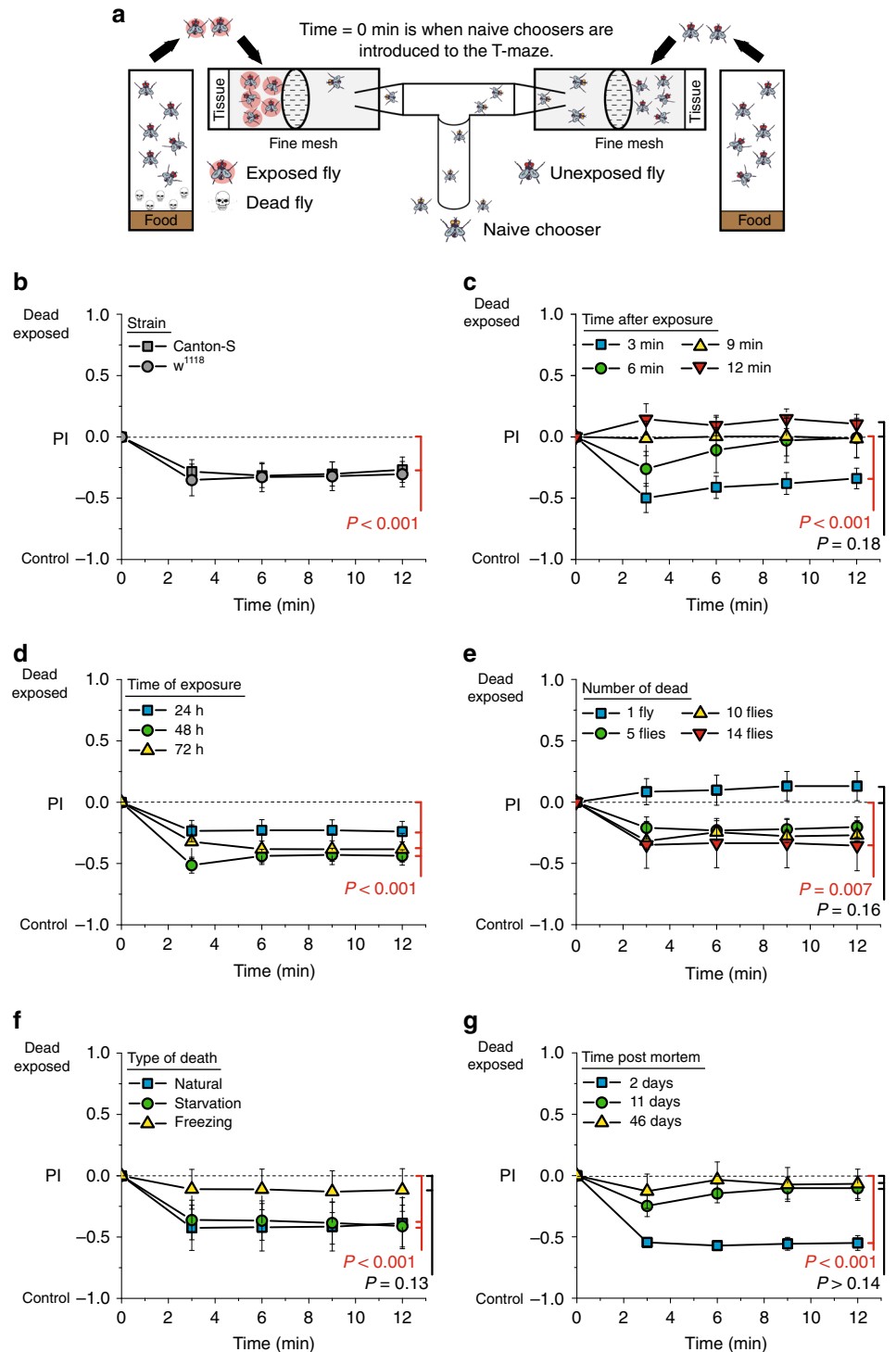

that exposure to dead animals from the two closely related species (*D. simulans* and *D. erecta*) were able to induce aversive cues in *D. melanogaster* to a similar extent as did exposure to their conspecifics, while exposure to the evolutionarily more distant *D. virilis* did not (Fig. 2a).

**Exposure to dead conspecifics alters physiology and lifespan.** Having observed that exposure of healthy flies to dead conspecifics consistently resulted in the production of aversive cues that repelled naive choosers, we next sought to investigate whether this treatment affected physiology and longevity in the exposed flies. We found that short-term exposure of *D.*

*melanogaster* to dead conspecifics compromised starvation survival and reduced levels of triacylglycerol (TAG), which is the primary storage lipid in flies (Fig. 2b, c). It also resulted in a moderate but significant reduction in $CO_2$ production, indicative of an altered metabolic rate (Fig. 2d). Exposed flies were capable of normal levels of spontaneous activity and exploration (Fig. 2e), but they showed impaired motivated climbing ability (Fig. 2f). Finally, chronic exposure to dead animals significantly reduced lifespan (Fig. 2g), which was robust to experimental strain (Supplementary Fig. 3a), was sex-specific in its magnitude (Supplementary Fig. 3b), was reduced in isolation (Supplementary Fig. 3c), and was not caused by population density

**Fig. 1** Flies become aversive after exposure to dead conspecifics. **a** Cartoon representing the exposure protocol and binary T-maze apparatus used in our choice behavior assays. PI = preference index calculated as (number of flies in the exposed arm ($N_E$) − number of flies in the unexposed arm ($N_C$))/total ($N_C + N_E$). **b** Flies of two different laboratory strains (Canton-S and $w^{1118}$) that were exposed to dead conspecifics for 48 h were aversive to naive Canton-S choosing females ($N = 9$ for Canton-S and $N = 7$ for $w^{1118}$, $P < 0.001$ for Canton-S and $P = 0.001$ for $w^{1118}$). **c** Flies exposed to dead conspecifics retained their aversive characteristics to naive choosing flies for up to 9 min after the dead flies were removed ($N = 9$ for each treatment, $P = 0.01$ for 6 min, and $P = 0.22$ for 9 min, group analysis of variance (ANOVA) $P < 0.001$). **d**, **e** When flies were exposed to dead conspecifics, they evoked avoidance behavior in naive choosing females that was intensified with **d** longer periods of exposure ($N = 19$ for 24 h, $N = 9$ for 48 h, and $N = 14$ for 72 h, group ANOVA $P = 0.027$) and **e** the number of dead animals used during the exposure treatment ($N = 6$ for each treatment, $P = 0.043$ for 5 flies, $P = 0.01$ for 10 flies, group ANOVA $P < 0.001$). **f** Flies exposed to animals that died of natural or starvation-induced death, but not freezing death, evoked avoidance behaviors in naive flies ($N = 8$ for natural and starvation-induced death and $N = 10$ for frozen induced death, group ANOVA $P = 0.04$). **g** Newly dead flies effectively induced the aversive cues in exposed animals, but long dead flies did not ($N = 6$ for each treatment, $P = 0.17$ for 11 days dead and $P = 0.14$ for 46 days dead flies, group ANOVA $P < 0.001$). Except where noted in **b**, all naive choosing female flies were from the Canton-S strain. Each T-maze sample tests 20 flies. Error bars represent standard error of the mean (SEM). All $P$ values were determined by non-parametric randomization (see Methods for details)

(Supplementary Fig. 3d) or by environmental structure (Supplementary Fig. 3e).

**Sight of dead is necessary and sufficient for aversive cues**. We hypothesized that the effects of exposure to dead animals relied on one or more sensory modalities in the healthy exposed flies. This is supported by the fact that gustatory and olfactory circuits have previously been shown to influence aging and physiology in *Drosophila*[8,9,27,28]. We therefore asked which sensory modalities were necessary for aversive cues to be triggered upon exposure to dead animals. We found that naive choosers exhibited no behavioral preference in the T-maze when the exposure to dead flies had taken place in the dark (Fig. 3a). To further demonstrate that the response is visually mediated and to assess potential interaction between light and dead flies, we repeated these experiments in our standard 12:12 h light–dark conditions using *norpA* mutant flies, which are blind. We observed no evidence of aversive cues from *norpA* mutants following exposure to dead flies (Fig. 3b).

*Orco*[2] mutant flies are broadly anosmic, and they exhibited a significant, but not complete, loss of the aversive cues (Fig. 3c). On the other hand, flies lacking the *ionotropic receptor 76b (Ir76b)*[29], which is involved in chemosensory detection of amino acids and salt, exhibited normal aversion following dead exposure, as did flies lacking both *Ir8a* and *Ir25a* ionotropic co-receptors, which are required for multiple sensory functions[30] (Supplementary Fig. 4a and b). Flies carrying a *poxn* mutant allele, which have impairment in taste perception, exhibited a similar response to death exposure as genetically homogenous control flies (Fig. 3d).

We next asked whether different sensory properties of the dead flies as stimuli were sufficient to induce aversiveness in healthy flies. Using a chamber that was designed to allow flies to see dead flies but remain physically separated from them (Supplementary Fig. 5a), we found that the sight of starvation-killed flies was sufficient to induced aversive cues to the same extent as direct exposure (Fig. 3e). Using this chamber, the sight of flies that had been killed by immersion in liquid nitrogen had no effect (Fig. 3e), thus replicating our earlier results that the type of death is important for the induction of aversive cues (Fig. 1f; Supplementary Fig. 6a), and that flies killed by freezing lack one or more key visual characteristics that are present in flies that died from starvation. The ability of flies to distinguish differences by sight may also explain why live *D. melanogaster* do not convey aversive signals when exposed to dead *D. virilis*, as these flies are darker and larger than *D. melanogaster* themselves (Supplementary Fig. 6b). Interestingly, however, repeated exposure to flies killed by freezing over a 20-day (but not 10-day) period was sufficient to induce aversiveness cues, indicating that flies may eventually learn to recognize these as dead or to respond to

alternative cues (Supplementary Fig. 5b, c). A second specialized chamber was used to investigate if olfactory cues (Supplementary Fig. 5d) were sufficient to induce aversion. An aversive effect was not seen in flies that were exposed to volatile odors from flies that died of starvation, indicating that olfactory cues alone were not sufficient to induce aversion (Fig. 3f). Olfactory cues from starved animals were also incapable of gating otherwise insufficient visual cues from animals that had died by freezing (Fig. 3g). Finally, aversiveness cues were not induced in flies by direct exposure to extracts from homogenized flies that had died by starvation, suggesting that gustatory cues are also not sufficient to induce aversiveness (Fig. 3h).

**Vision is necessary for impaired lifespan in exposed flies**. The significant reduction in lifespan that resulted from chronic exposure to dead animals was absent when flies were aged in constant darkness (Fig. 4a; Supplementary Fig. 7a). Unexposed control flies were longer-lived in constant darkness, which is consistent with an effect of death perception in cohorts aging normally in light–dark cycles. Blind *norpA* mutants also exhibited significantly reduced effects of exposure to dead flies on lifespan (Fig. 4b; Supplementary Fig. 7c). Long-term exposure to flies killed by freezing reduced lifespan (Supplementary Fig. 7b). Largely anosmic *Orco*[2] mutant flies exhibited a partial decrease of the effect of exposure to dead flies on lifespan (Fig. 4c), while flies lacking *Ir76b* or flies lacking both *Ir8a* and *Ir25a* exhibited a decreased lifespan following exposure to dead flies to the same extent as genetically homogenous control animals (Supplementary Fig. 4c and d). Finally, *poxn* mutants also exhibited a similar decrease in lifespan in response to exposure to dead flies as control flies (Fig. 4d; Supplementary Fig. 7c).

Our findings suggest the effects of exposure to dead animals are not caused by infection, bacterial proliferation, or changes in the gut biota in either dead or exposed animals. Similar levels of aversion were induced when axenic flies were used as both exposed and dead flies, establishing that these factors do not play a significant role in the aversive response (Supplementary Fig. 8). It remains possible that the effects of death exposure on aversion and lifespan are, at least in part, mechanistically distinct, and we were unable to formally test the latter because lifespan exposure experiments comprised entirely of axenic animals is technically unfeasible. To date, we know of no evidence that the disparate environmental and genetic manipulations used here to reduce or eliminate the effects of exposure to dead conspecifics on lifespan affect the microbiota, but nonetheless we cannot rule this out as a potential confound.

Together, these data indicate that there is an essential perceptual component associated with the physiological and health effects of exposure to dead conspecifics. Sight of naturally dead conspecifics is both necessary and sufficient to induce

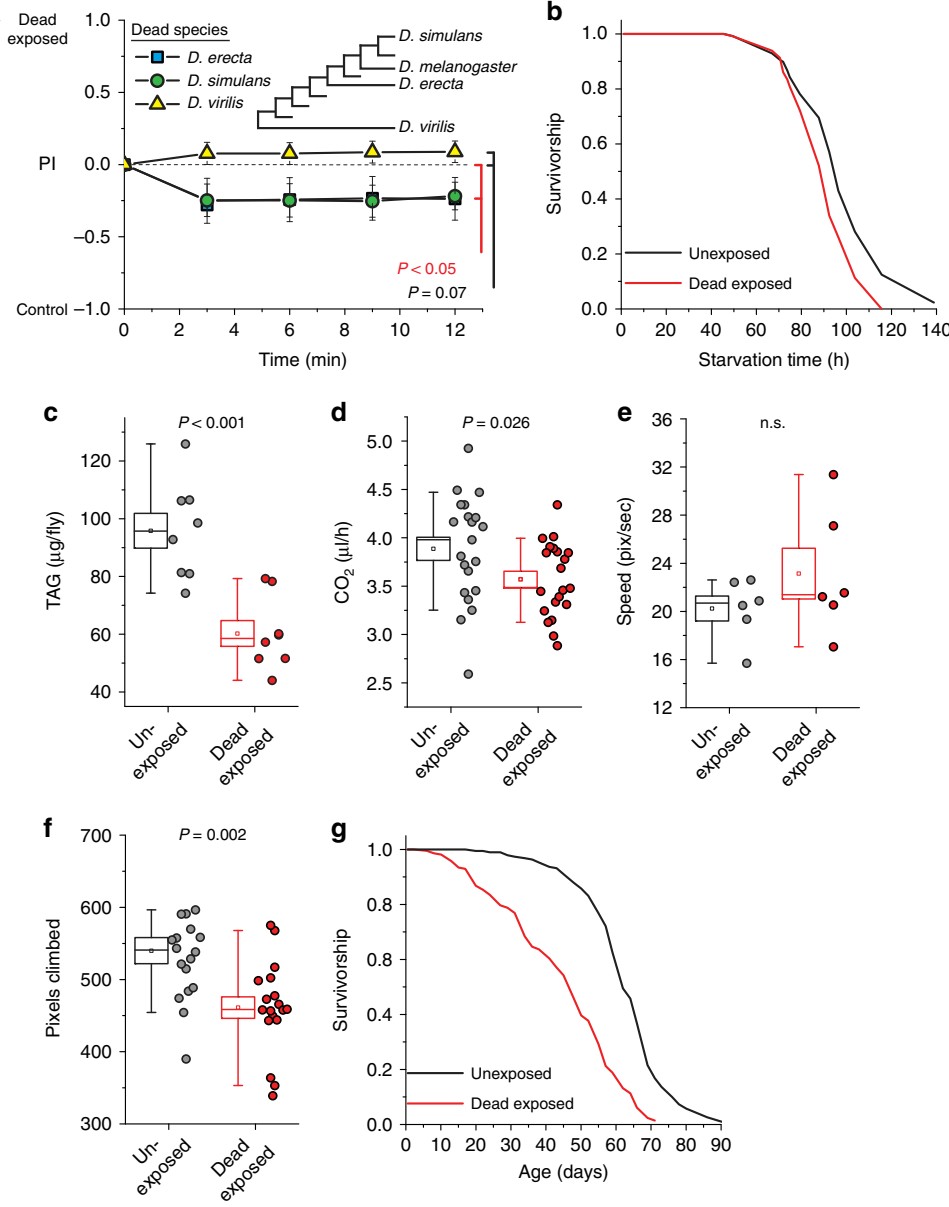

**Fig. 2** Exposure to dead conspecifics alters physiology and lifespan in *Drosophila melanogaster*. **a** When *D. melanogaster* were exposed to dead animals from each of two closely related species (*D. simulans* and *D. erecta*), they presented aversive cues, but exposure to the evolutionarily more distant *D. virilis* had no effect (N = 10 for *D. erecta*, N = 8 for *D. simulans*, and N = 17 for *D. virilis*, P = 0.046 for *D. erecta* and P = 0.008 for *D. simulans*, group analysis of variance (ANOVA) P < 0.001). Inset depicts a phylogeny of related *Drosophila* species. **b–d** Flies exposed to dead conspecifics exhibited reduced **b** starvation survival (N = 100 per treatment, P < 0.001), **c** triacylglyceride stores (TAG, N = 8 biological replicates of 10 flies each), and **d** metabolic rate as measured by $CO_2$ production relative to unexposed animals (N = 21 biological replicates of 5 flies/treatment). **e, f** While exposure to dead conspecifics did not affect **e** spontaneous movement rates (N = 6 for each treatment, P = 0.26), **f** forced climbing was impaired relative to unexposed animals (N = 18 for each treatment). **g** Chronic exposure to dead animals significantly reduced lifespan flies (N = 190 for unexposed, 212 for exposed, P < 0.001). For **a**, naive choosing flies were from the Canton-S strain. Each T-maze sample tests 20 flies. Error bars represent standard error of the mean (SEM). P values for binary choice were determined by non-parametric randomization. Comparison of survival curves was via log-rank test, and the remaining phenotypes were evaluated for significance by t test (see Methods for details)

physiological effects, suggesting a model in which visual cues serve as the primary way in which *D. melanogaster* distinguish dead flies. While gustatory cues are not involved in the effects that we observe, the role of olfaction is less clear. Smell-deficient flies respond less strongly to dead conspecifics, but odors from dead flies are not sufficient to induce changes in aversiveness. The aversive cues emitted by flies following exposure to dead individuals do have a significant olfactory component: when *Orco²* mutants were used as naive choosers in the T-maze assay (e.g., described in Fig. 1a), they assorted randomly between the

two arms (Supplementary Fig. 9). Similar results were observed when naive choosers carried a loss of function mutation for *Gr63a*, an essential component of the *Drosophila* $CO_2$ receptor (Supplementary Fig. 9). We therefore currently favor a model in which olfaction mediates a social cue among exposed animals as a result of visual perception of dead flies.

**Changes in the head metabolome follow exposure to dead flies.** We next asked whether we might identify a signature of this putative perceptive event by comparing the metabolome of the

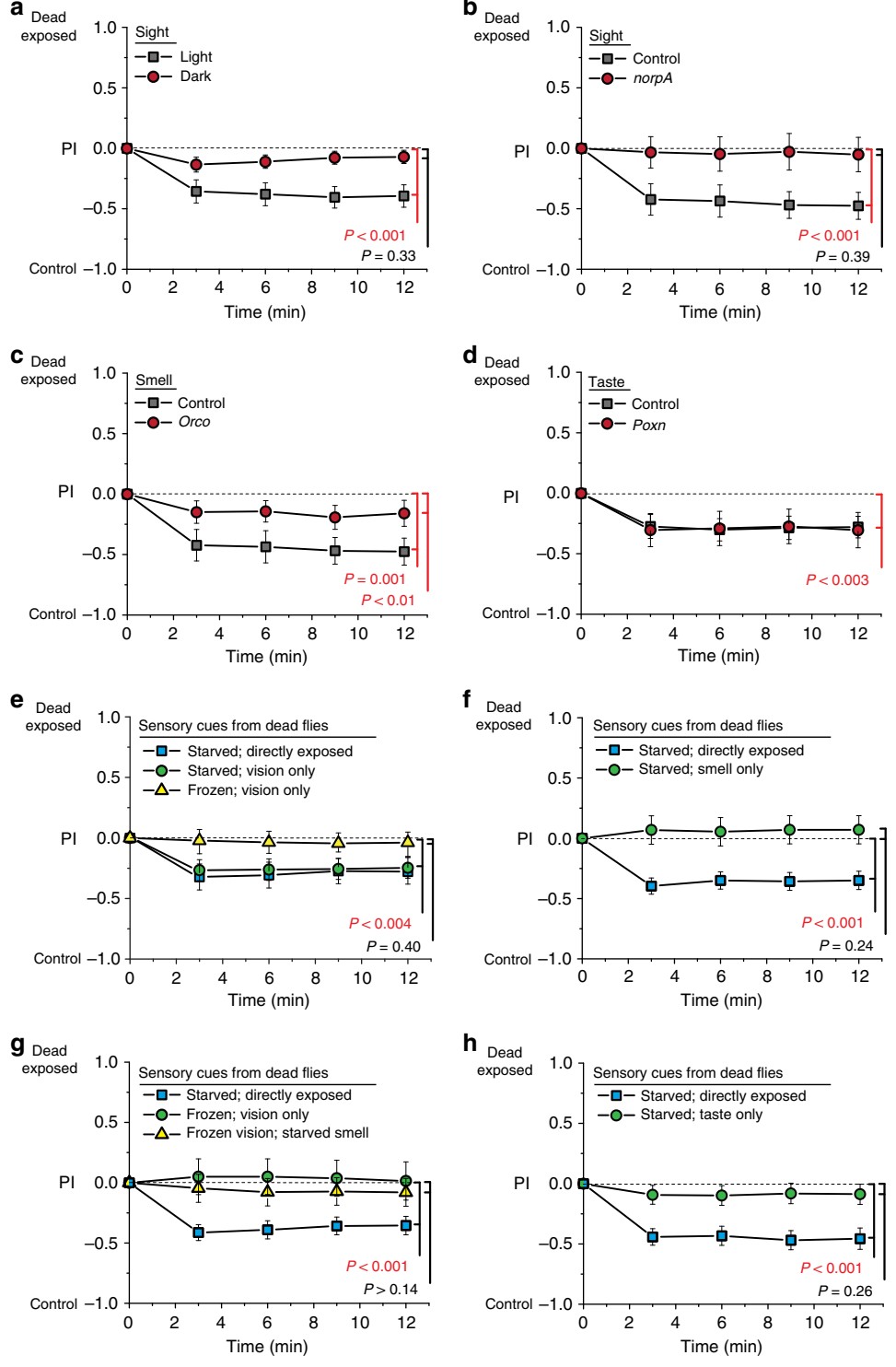

homogenized heads of experimental flies following 48 h of dead exposure to that of unexposed animals. *norpA* mutant flies, which were treated identically but which lacked the ability to see dead flies, were analyzed simultaneously to account for temporal effects and to isolate potential causal metabolites. Targeted metabolite analyses identified 119 metabolites present in treatments for positive and negative modes (see Supplementary file for raw data). Using a randomization procedure together with principal component analysis (PCA), we identified a single principal component (PC10) that significantly distinguished the neuro-metabolomes of exposed and unexposed flies but was unchanged

by exposure in *norpA* mutant flies (Fig. 5a; Supplementary Fig. 10a). The multivariate analysis revealed that five of the top ten metabolites associated with the effects of exposure to dead flies (i.e., those strongly loaded on PC10) have been implicated in models of anxiety, depression, and/or mood disorders in mammals (specifically lactate[31], quinolinate[32], sorbitol[33], 3-hydroxybutyric acid[34], and sarcosine[35]). In addition to our systems biology analysis, we also asked whether any individual metabolites were statistically correlated with exposure to dead individuals in experimental but not *norpA* mutant flies (i.e., exhibited a statistically significant interaction between

**Fig. 3** Sensory perception is required for the induction of aversive cues following exposure to dead flies. **a** When flies were exposed to dead flies in the dark, they failed to evoke avoidance behavior in naive choosing females ($N = 19$ for light exposure and $N = 20$ for dark exposure). **b** Blind, *norpA* mutant flies exposed under lighted conditions also failed to induce aversive cues following exposure to dead flies ($N = 8$ for each treatment). **c** *Orco²* mutant flies, which have impaired olfaction, evoked a small, but significant, avoidance behavior in choosing females following death exposure ($N = 9$ for *Orco²* and $N = 8$ for control). **d** Flies carrying the *Poxn^{ΔM22-B5-ΔXB}* mutation, which have impaired taste function, exhibited a similar induction of aversive cues in response to death exposure as did control flies ($N = 9$ for each treatment, $P = 0.002$ for control and $P < 0.001$ for *poxn*). **e** The sight of starvation-killed flies was sufficient to induce aversive cues to the same extent as direct exposure, while the sight of flies killed by immersion in liquid nitrogen had no effect ($N = 19$ for direct exposure, $N = 20$ for vision only). **f** The smell of starvation-killed flies, which was provided by isolating dead animals behind a fine mesh screen, failed to induce aversive cues ($N = 10$ for each treatment). **g** The sight of flies killed by freezing in liquid nitrogen failed to induce aversive cues (frozen; vision only), and this was not affected by simultaneous smell of starvation-killed animals ($N = 8$ for frozen only, $N = 13$ for starved and starved + frozen). **h** Homogenized dead flies failed to evoke avoidance behavior in naive flies ($N = 10$ for ground up and control treatments). For binary choice assays, all exposed flies and naive choosing flies were from the Canton-S strain. Each T-maze sample tests 20 flies. Error bars represent standard error of the mean (SEM). *P* values for binary choice were determined by non-parametric randomization

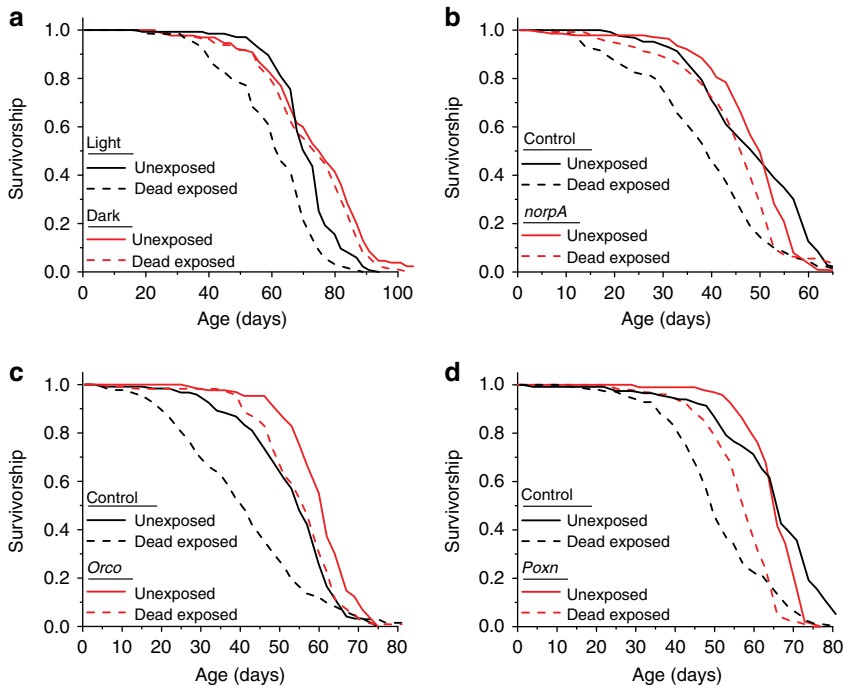

**Fig. 4** Exposure of dead conspecifics causes changes in lifespan that are mediated by sight and smell. **a** Exposure of live animals to dead in the light, but not the dark, affected lifespan ($N = 118$ for dark dead exposed, $N = 130$ for dark unexposed, $N = 125$ for light dead exposed, and $N = 135$ for light unexposed, $P < 0.001$ for light and $P = 0.18$ for dark, $P < 0.001$ for the interaction between light and exposure via Cox regression). **b** The lifespan effect observed in *norpA* mutant flies following exposure to dead conspecifics was significantly diminished relative to control animals ($N = 139$ for *norpA* dead exposed, $N = 126$ for *norpA* unexposed, $N = 126$ for control dead exposed, and $N = 131$ for control unexposed, $P < 0.001$ for control and $P = 0.003$ for *norpA*, $P < 0.001$ for the interaction between genotype and exposure via Cox regression). **c** Anosmic flies maintained a reduced effect of death exposure on lifespan compared to control flies ($N = 115$ for *Orco²* dead exposed, $N = 129$ for *Orco²* unexposed, $N = 133$ for control dead exposed, and $N = 121$ for control unexposed, $P < 0.001$ for control and *Orco²*, $P = 0.002$ for the interaction between genotype and exposure via analysis of variance (ANOVA)). **d** Taste-blind flies showed normal lifespan effects due to dead exposure ($N = 101$ or *Poxn* dead exposed, $N = 96$ for *Poxn* unexposed, $N = 112$ for control dead exposed, and $N = 116$ for control unexposed, $P < 0.001$ for control and *Poxn*, $P = 0.06$ for the interaction between genotype and exposure via Cox regression). Survival curve comparison was accomplished using a log-rank test (see Methods for details)

treatments). We found only one, glyceraldehyde (Fig. 5b), likely due to limited statistical power for this question.

**Serotonin signaling mediates the effects of exposure to dead.** Following the metabolomic analysis, we focused on pharmacologic compounds with anti-depressant or anti-anxiety effects with the goal of identifying molecular targets that are required for the health consequences of exposure to dead flies (Supplementary Table 1). For T-maze assays, flies were treated with each drug prior to exposure to dead individuals, while for lifespan assays flies were treated throughout life. Of those compounds examined, we found one, pirenperone, which abrogated the effects of death perception on both aversive cues and lifespan (Fig. 5c, d;

Supplementary Table 1). Pirenperone is a putative antagonist of the serotonin 5-HT2 receptor in mammals, although it may interact with other biogenic amine receptors at high concentrations[36]. We therefore tested whether loss of the 5-HT2A receptor recapitulated the effects of pirenperone and abrogated the effects of death perception on aversion and lifespan in *Drosophila*. We found that it did (Fig. 5e, f; Supplementary Fig. 10b, c), suggesting that serotonin signaling through the 5-HT2A branch is required to modulate health and lifespan in response to this perceptive experience.

Finally, we asked whether activation of 5-HT2A⁺ neurons was sufficient to recapitulate the aversion and lifespan phenotypes we observed following death perception. We ectopically expressed

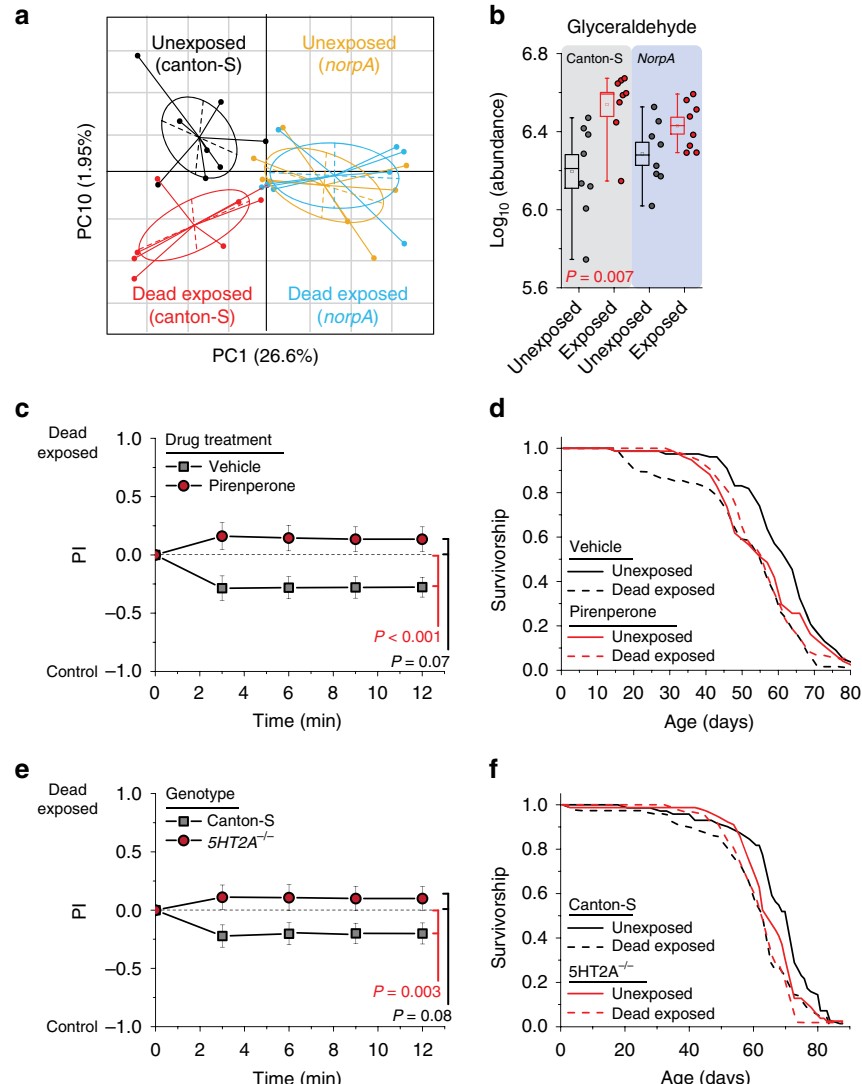

**Fig. 5** Death perception elicits acute changes in the neuro-metabolome, and its effects on health are mitigated by manipulations that attenuate serotonin signaling. **a** Principal component plot showing the distribution of samples for each treatment. Neuro-metabolites weighted heavily in PC10 distinguish the effect of death exposure, while those favored in PC1 distinguish genotype. Plots represent mass spectrometry analysis of metabolites identified under positive mode ($N = 8$ biological replicates, with 40 fly heads per replicate). **b** Glyceraldehyde abundance was significantly increased in flies following death perception, but it was unchanged in blind *norpA* mutant flies similarly treated. **c**, **d** Pharmacologic treatment of Canton-S females with the serotonin receptor 5-HT2 antagonist, pirenperone, during exposure to dead conspecifics effectively protected them from the consequences of death perception on **c** aversive cues detected by naive choosing flies ($N = 10$ for each treatment) and **d** lifespan ($N = 75$ for pirenperone-fed dead exposed, $N = 76$ for pirenperone-fed unexposed, $N = 75$ for vehicle-fed dead exposed, and $N = 77$ for vehicle-fed unexposed, $P < 0.001$ for vehicle-fed and $P = 0.86$ for pirenperone-fed, $P = 0.007$ for the interaction between drug and exposure via analysis of variance (ANOVA)). **e**, **f** Null mutation of serotonin receptor 5-HT2A-protected flies from the consequences of death perception on **e** aversive cues detected by naive choosing flies ($N = 15$ for each treatment) and **f** lifespan ($N = 63$ for $5\text{-}HT2A^{-/-}$ dead exposed, $N = 78$ for $5\text{-}HT2A^{-/-}$ unexposed, $N = 75$ for Canton-S dead exposed, and $N = 72$ for Canton-S unexposed, $P < 0.001$ for Canton-S control flies and $P = 0.06$ for $5\text{-}HT2A^{-/-}$ mutants). A replicate lifespan experiment revealed the same results (see Supplementary Fig. 5c), and $P = 0.05$ for the combined interaction between genotype and exposure via ANOVA. $P$ values for principal component analysis and for binary choice were determined by non-parametric randomization. Each T-maze sample tests 20 flies. Error bars represent standard error of the mean (SEM). Comparison of survival curves was via log-rank test, and individual metabolites were evaluated for significance by $t$ test (see Methods for details)

the thermosensitive cation channel transient receptor potential A1 (TRPA1) in cells that putatively produce 5-HT2A (using *5-HT2A-GAL4*). The *Drosophila* TrpA1 channel promotes neuron depolarization only at elevated temperatures (>25 °C), thereby allowing temporal control over cell activation[37]. Transgenic flies (*5-HT2A-GAL4>UAS-TrpA1*) and their genetic control strain (*5-HT2A-GAL4;+*) were raised and maintained for 10 days after eclosion at 18 °C (non-activating conditions). They were then placed at 29 °C for 48 h to activate the neurons, over a time frame similar to the death perception assays. We found that after

activation, transgenic flies were aversive relative to the genetic control (Fig. 6a; $P < 0.001$). Chronic activation of $5\text{-}HT2A^+$ neurons throughout life also decreased longevity consistent with our previous data (Fig. 6b).

## Discussion

Sensory perception has emerged as a potent modulator of aging and physiology across taxa. Exposure of *Drosophila* and *C. elegans* to food-based odorants limits the beneficial effects of dietary restriction[8,38], while perception of the opposite sex modulates

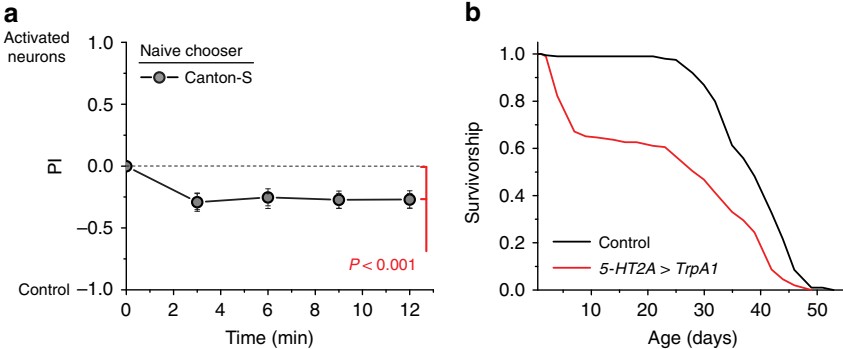

**Fig. 6** Activation of 5-HT2A neurons induces aversiveness and reduces lifespan in *Drosophila*. Constitutive activation of 5-HT2A+ neurons via expression of *UAS-TrpA1* relative to *5-HT2A-GAL4;+* control flies results in **a** increased aversiveness ($N = 19$, $P < 0.001$) and **b** significantly decreased lifespan ($N = 194$ for *5-HT2A>TrpA1* and $N = 195$ for *5-HT2A-GAL4;+*, $P < 0.001$). Each T-maze sample tests 20 flies. Error bars represent standard error of the mean (SEM). Comparison of survival curves was via log-rank test

lifespan through neural circuits that utilize conserved neuropeptides to establish motivation and reward[9,10,13]. Loss of specific olfactory and gustatory neurons modulates lifespan and physiology and influences measures of healthy aging, including sleep and daily activity patterns[6,28,39,40]. Our data are consistent with an additional perceptive influence that affects longevity in *Drosophila*, exposure to dead conspecifics, which depends on visual and olfactory function. Not all dead flies induced such effects (flies that died by freezing or flies that were long dead or were of a distantly related species failed to do so), suggesting that flies have the perceptive ability to distinguish differences in these carcasses. These results are consistent with reports documenting sufficient visual acuity in flies to distinguish different ecologically relevant cues in their environment, such as parasites and competitors[41–43]. Additional experiments designed to identify the precise cues that convey information about dead conspecifics to exposed flies, and onto non-exposed counterparts, will be required.

This report adds to growing evidence that serotonin is an important component of how different sensory experiences modulate aging and aging-related disease across taxa. Serotonin modulates sensory integration in mammals[44] and has been linked to the longevity effects associated with sensory perception of food[11,45] and hypoxia[46–48]. In *C. elegans*, a global, cell non-autonomous response to heat is triggered by thermo-sensory neurons in a serotonin-dependent manner, which is also capable of extending lifespan[49]. Lifespan extension by activation of the hypoxic response also requires specific components of serotonin signaling in sensory neurons[46]. In *Drosophila*, serotonin is required for protein perception, and loss of receptor 5-HT2A increases fly lifespan in complex and potentially stressful nutritional environments[11]. Whether serotonin is merely permissive for changes that affect lifespan or whether it directly modulates aging remains unclear. There is evidence for a direct role of *C. elegans*, where feeding worms serotonin receptor antagonists is sufficient to extend lifespan by putatively mimicking dietary restriction[48]. We present evidence that activation of 5-HT2A neurons is sufficient to modulate aging, but these results should be interpreted with caution; both aversiveness and short lifespan might instead be reflective of unrelated molecular changes that result in less healthy animals.

Although death perception is known to occur in several species throughout the animal kingdom[18], this is, to our knowledge, the first indication that such an ability may be present in an invertebrate laboratory model system. We suggest a model in which dead conspecifics serve as a threat cue that results in negative consequences on metabolism, physical condition, and aging[50,51]. Neural states akin to fear, anxiety, and depression have recently been described in *Drosophila*, with phenotypic manifestations and molecular causes that are consistent with those observed in mammals[51–54]. Moreover, associations linking human morbidity and aging-related disease are prevalent in the demographic and epidemiological literatures[55–57]. Invertebrate models systems allow us to study the mechanisms underlying how aging and other health metrics are determined by neuronal circuits that emanate from sensory tissues and that influence conserved neuroendocrine processes. These may have evolved to modulate physiology in response to the perception of environmental conditions that affect evolutionary fitness, such as the presence of food, mates, or danger[58].

## Methods

**Contact for reagent and resource sharing**. Further information and requests for resources and reagents should be directed to and will be fulfilled by the Lead Contact, Scott D. Pletcher (spletch@umich.edu).

**Experimental model and subject details**. The laboratory stocks $w^{1118}$, Canton-S, *UAS-dTrpA1*, *norpA*, and *Ir76b*$^{−/−}$ [BL51309] *Drosophila* lines were obtained from the Bloomington Stock Center. *Poxn*$^{ΔM22−B5ΔXB}$ and *Poxn*$^{Full1}$ were provided by J. Alcedo[59]. *Orco*$^{2}$ mutant flies were a generous gift from L. Vosshall[60]. *Gr63a*$^{1}$ mutant flies were a gift from A. Ray. *5-HT2A*$^{PL00052}$ mutant and *5-HT2A-GAL4* (*3299-GAL4*) flies were graciously provided by H. Dierick. *Ir8a*$^{−/−}$/*Ir25a*$^{−/−}$ mutant flies were kindly provided by R. Benton. Three species of *Drosophila* (*D. simulans*, *D. erecta*, and *D. virilis*) were generously provided by P. Wittkopp. All of these strains were maintained on standard food at 25 °C and 60% relative humidity in a 12:12 h light–dark cycle.

**Generation of dead flies**. Unless otherwise noted, dead flies were generated by starvation. One- to two-week-old Canton-S female flies were separated using $CO_2$ anesthesia and transferred to vials containing 2% agar. Flies were transferred to fresh agar vials every ~3 days and dead flies were collected within 3 days of death. Vials in which dead flies stuck to the agar were not used. For the data in Fig. 1f, flies died from natural causes or were killed by rapid freezing in liquid nitrogen. Age-matched Canton-S females were used for rapid freezing.

**Short-term exposure to dead flies**. Twenty 2-week-old mated female flies were collected under light $CO_2$ anesthesia and exposed for 48 h to 14 freshly dead female flies in standard food, where they freely interacted with the dead flies. During the 48h exposure period, flies were maintained in a 12:12 h light–dark cycle, except for those involving exposure in 100% darkness, which took place in a closed incubator. In both cases, flies were maintained at 25 °C and 60% relative humidity. To test species specificity, experimental female flies (*D. melanogaster*) were exposed to three different species of dead *Drosophila*; we used 14 freshly dead female flies of *D. melanogaster*, *D. simulans*, and *D. erecta*, and 8 freshly dead *D. virilis* due to their larger size.

**Behavioral preference assays**. To generate naive choosers for preference assays, newly eclosed virgin female flies (<7 h old) were collected and transferred to standard food vials with 3 male flies per 20 females. Flies were kept at 25 °C and 60% humidity, with a 12:12 h light–dark cycle for 2 days, after which they were briefly anesthetized to

remove the male flies. Mated female flies were then transferred to fresh vials for one day to recover. On day 4 post eclosion, the flies were placed into vials containing moist tissue paper for 4 h prior to their introduction into the T-maze for behavioral monitoring. Choice was measured using binary traps made from commercially available T connectors (McMaster-Carr Part Number 5372K615) with 200 µl pipette tips, which were trimmed, attached to opposite ends of the T connectors to form one-way doors that end in small collection chambers. Experimental flies that were pre-exposed to dead flies were loaded into a collection chamber with moist tissue paper in one arm of the T-connector, while unexposed flies were loaded into the opposite chamber that also contained moist tissue paper. Unless otherwise noted, 20 live flies were exposed to 14 dead animals. Modifications were made to the apparatus to test specific sensory modalities:

*Vision only*: Canton-S mated female flies were exposed to flies that were dead either due to starvation or to liquid nitrogen immersion where the dead were kept beneath an acrylic floor on a thin layer of agar. This apparatus ensures that only visual cues are transferred. As a control treatment, a group of flies was directly exposed to starvation-induced dead for 48 h in the absence of the acrylic barrier.

*Olfaction only*: Mated Canton-S female flies were co-housed with starvation-killed flies, the latter of which were physically separated from exposed animals by the presence of a fine mesh screen that prevented physical interaction between the live and dead flies. As a control treatment, no flies were loaded behind the screen.

*Gustatory only*: Starvation-induced dead flies were homogenized and spread on top of the food prior to exposure. Mated Canton-S female flies were then exposed to either intact or grounded dead flies for 2 days.

Twenty naive choosing flies were introduced into the central arm of the maze, and the number of flies trapped in each arm was counted at regular intervals. Behavioral preference was measured in a dark room under dim 660 nm red light at 24 °C, and behavior was observed at 3, 6, 9, and 12 min. A preference index (PI) at each time point was computed as follows: (number of flies in exposed arm ($N_E$) − number of flies in unexposed arms ($N_C$))/($N_C + N_E$). The fraction of flies that participated in the experiments was calculated as: ($N_C + N_E$)/20. Average PI values are weighted mean values among replicates with weights proportional to the number of animals that made a choice. Participation rates for all of the T-maze assays were >50%. Experiments were replicated at least two independent times. Beads used for mock dead flies were obtained from Cospheric innovation in Microtechnology (Catalog number: CAS-BK 1.5 mm).

**Starvation experiments**. At day 4 post eclosion, 10 mated female flies were separated in 10 vials/treatment containing 2% agar. Flies were kept in constant temperature and humidity conditions with a 12:12 h light–dark cycle. A census of live flies was taken every 6–8 h. For exposed flies, the dead flies were left in each vial throughout the experiment. For control flies, dead flies were removed at each census point. Flies were transferred to fresh agar vials every 6 h.

**TAG assays**. Four-day-old, adult Canton-S female flies were collected and subsequently handled using our standard short-term exposure protocol (see above). Following the 48 h exposure to dead animals, live experimental flies were removed and homogenized in groups of 10 in 150 µl phosphate-buffered saline (PBS)/0.05% Triton X. Unexposed flies were collected simultaneously. The amount of TAG in each sample was measured using the Infinity Triglyceride reagent (Thermo Electron Corp.) according to the manufacturer's instructions. Eight independent biological replicates (of 10 flies each) were obtained for treatment and control cohorts.

**Negative geotaxis assay**. Four-day-old, adult Canton-S female flies were collected and subsequently subjected to our standard short-term exposure protocol (see above). Following the 48 h exposure to dead animals, live experimental flies and their corresponding unexposed controls were removed and transferred to climbing chambers by aspiration. Negative geotaxis was measured by DDrop, an automated machine developed in the Pletcher laboratory that drops flies from 24 in. and then tracks upward movement of individual flies through a video tracking algorithm. For each fly, we calculated both the total distance traveled and the time required to reach individual quadrants of the chamber.

**CO₂ measurement**. Four-day-old, adult Canton-S female flies were collected and subsequently subjected to our standard short-term exposure protocol (see above). Following the 48 h exposure to dead animals, live experimental flies and corresponding unexposed controls were removed, and $CO_2$ production was measured from groups of five female experimental flies alongside their corresponding unexposed controls at 25 °C. We used a Sable Systems Respirometry System, including a LiCor LI-7000 carbon dioxide analyzer, a Mass Flow Controllers (MFC2), and a UI-2 analog signal unit. Immediately prior to analysis, flies were transferred without anesthesia into glass, cylindrical respirometry chambers. Flies were allowed to acclimate to the new environment for 8 min before $CO_2$ collection began. Six chambers were analyzed simultaneously using stop-flow analysis and the Sable Systems multiplexer. Incoming atmospheric air flow was dried, scrubbed of $CO_2$, and then rehydrated before entering the respirometry chambers via the multiplexer. For each group, we collected three measures of $CO_2$ production over a period of 20 min that were averaged to determine a final, single estimate of $CO_2$

production per group. $CO_2$ production values were obtained using the EXPDATA software from Sable Systems, following adjustment using a proportional baseline.

**Video tracking**. Mated Canton-S female flies were starved for 4 h in a vial with moist tissue paper prior to the experiment to encourage movement within the chambers. Each chamber contained two dumbbell-shaped arenas comprised of two circles (internal diameter = 1.0 in.) separated by a narrow corridor connecting them. A thin 2% layer of agar served as the floor of the chambers. Dead flies were lightly pressed into the agar of 1 arena to secure them. The positions of these stimuli were randomized within each experiment. Five minutes prior to recording, single exposed flies were loaded into the chambers by aspiration. Movement in each arena was recorded for 2 h in a 25 °C incubator under white light. Recordings were analyzed using the DTrack Software, developed in the Pletcher laboratory[6,61]. From the tracking data, we calculated the speed, total distance traveled, and amount of time each fly spent in each side of the arena.

**Feeding analysis**. Feeding behavior was measured using the fly liquid interaction counter (FLIC) as described previously[62]. Following our standard 48 h exposure treatment, individual Canton-S female flies were placed into a single FLIC chamber with two food wells, each containing a 10% sucrose solution. Two independent experimental blocks were conducted using 15 dead exposed and 15 unexposed flies per experiment, providing a total of 30 flies per treatment. The experiments were performed at constant temperature (25 °C) with 12:12 h light–dark cycle. Throughout the experiment, three dead flies were kept in the chambers assigned to the dead exposed condition. Feeding interactions with the food were measured for 24 h continuously using the FLIC reservoir system (see http://www.wikiflic.com). Data were analyzed using the FLIC Analysis R Source Code (available from wikiflic.com). Relevant feeding measures included the number of total interactions with the food, the total time spent interacting with the food, mean duration of each putative feeding event, and mean time between feeding events. These data were determined to be normally distributed, and a *t* test was used to determine whether statistically significant differences were observed after noting the absence of significant block effects.

**Axenic fly culturing**. Canton-S flies were placed into cages with purple grape agar and yeast paste for approximately 18 h. Embryos were then collected with 10 ml PBS and moved into a sterile hood, where they were treated with 10 ml 1:10 sterile bleach solution (3× washes) and then washed with sterile water. Using sterile technique, 8 µl of embryos were aliquoted into sterile 50 ml falcon tubes that contained 8 ml sterile standard fly media. Embryos were allowed to develop in a humidified incubator at 25 °C with 12:12 h light–dark cycles. After 12 days, axenic flies were collected in a sterile hood into fresh, sterile 50 ml falcon tubes containing 8 ml sterile standard fly media. Flies were aged for 2 weeks, during which time fresh, sterile media were provided every 2–3 days. After 2 weeks, the flies were split into two groups, with half of the group given fresh, sterile agar (to generate sterile dead flies) and the other half given fresh, sterile standard fly media. Sterile dead exposures occurred in the sterile hood using sterile tools and sterile technique. Control, conventionally reared flies (non-sterile) were handled in an identical manner, with the exception of bleach washing. The sterility of the axenic flies was verified by plating the supernatant from total fly extracts onto brain heart infusion agar plates; colonies grew from the extracts of traditionally reared flies but never from the extracts of sterile flies.

**Metabolomic analysis**. Following 48 h of dead fly exposure, experimental flies were quickly frozen in a dry ice bath, and stored at −80 °C overnight. Heads were removed via vortexing and manually separated from the body parts. Forty heads were then homogenized for 20 s in 200 µl of a 1:4 (v:v) water–MeOH solvent mixture using the Fast Prep 24 (MP Biomedicals). Following the addition of 800 µl of methanol, the samples were incubated for 30 min on dry ice, and then homogenized again. The mixture was spun at 13,000 RPM for 5 min at 4 °C, and the soluble extract was collected into vials. This extract was then dried in a speedvac at 30 °C for approximately 3 h. Using a LC-QQQ-MS machine in the MRM mode, we targeted ~200 metabolites in 25 important metabolic pathways, in both positive and negative MS modes. After removing any metabolites missing from more than 5 out of 32 samples (15%), we were left with 119 metabolites. Metabolite abundance for remaining missing values in this data set were log-transformed and imputed using the *k*-nearest neighbor algorithm with the impute package of R Bioconductor (www.bioconductor.org). We then normalized the data to the standard normal distribution ($\mu = 0$, $\sigma2 = 1$). PCA was performed using the made4 package of R Bioconductor. We used permutation tests ($n = 10,000$) to select PCs that significantly separate between different treatments (genotype and/or exposure to dead flies). For each permutation, we randomly distributed the treatments to the real abundance of each metabolite. PC analysis was done for both randomized and real data. The degree of separation for each PC can be measured by analyzing between- and within-group variance based on the projection of samples on that PC, which is indicated by the Z-score:

$$Z = \frac{\text{Variance between groups}}{\text{Variance within groups}}.$$

Variance between groups $= \sum_{k=1}^{N} n_k \times (\text{Mean}_{\text{total}} - \text{Mean}_k)^2$, where $N$ indicates

the number of groups and $n_k$ indicates the number of samples in group $k$. The distribution of $Z$-score was obtained from 10,000 randomized datasets. PCs that significantly deviated from this randomized distribution were considered as a significant separation of groups.

To identify individual metabolites of interest that are likely to be associated with death exposure, we sorted them per loading on PC10 and selected the top 10. For these candidates, we looked for metabolites that were (i) significantly different between control flies exposed to dead conspecifics vs. those not exposed and that were (ii) less affected in *norpA* mutant flies. We found glyceraldehyde significantly up-regulated upon death exposure in Canton-S flies (one-sided Student's $t$ test, $P = 0.007$), whereas such differences were not significant in *norpA* flies (one-sided Student's $t$ test, $P = 0.07$).

**Survival experiments.** For lifespan experiments, experimental and control flies were reared under controlled larval density and collected as adults within 24 h of emergence onto standard food where they were allowed to mate freely for 2–3 days. At 3 days post eclosion, female flies were sorted under light $CO_2$ and placed into fresh food vials. Contrary to our short-term protocol, experimental flies were chronically exposed to dead animals throughout their life. Lifespan measures were obtained using well-established protocols[63]. Flies were transferred to fresh food vials every Monday, Wednesday, and Friday, at which time 14 freshly dead flies (~2–3 days old) were added. Unexposed animals were transferred simultaneously, but instead of adding dead flies we removed any flies that had died since the last census time. Flies were maintained at 25 °C and 60% humidity under a 12:12 h light–dark cycle. For experiments in the dark, flies were maintained in a dark incubator at 25 °C and 50% humidity. Vials were changed as described above, and dead flies were counted under dim red light.

**Drug administration.** All drugs were purchased from Sigma-Aldrich. Each drug was initially dissolved in 100% dimethyl sulfoxide (DMSO) at 10 mM concentration, aliquoted, and stored at −20 °C. Every Monday, Wednesday, and Friday, an aliquot of the drug stock was thawed and diluted 1:500 in water for lifespans (20 μM final concentration). A similar dilution of DMSO alone was made in water as a vehicle control. Then, 100 μl of the diluted drug or vehicle control was added to each vial, coating the top of the food surface. After the liquid evaporated (~2 h), the vials were ready for use as described above. For the behavior assay, flies were pretreated with 1 mM pirenperone or an equivalent dilution of DMSO for 2 weeks prior to exposure, in the absence of dead flies.

**5-HT2A⁺ neuron activation methods.** *5-HT2A-GAL4* (*3299-GAL4*, H. Dierick, BCM) flies were crossed to *UAS-dTrpA1* to generate *5-HT2A>dTrpA1* flies. *UAS-dTrpA1* was backcrossed for at least 10 generations to $w^{1118}$ and therefore *5-HT2A-GAL4;+* was used as a genetic control strain. For behavioral assays, progeny from all crosses were maintained at 18 °C until they were 10–14 days old, after which they were transferred to 29 °C for 48 h prior to mimic exposure to dead individuals. The T-maze assay was established with *5-HT2A>dTrpA1* on one side of the assay and *5-HT2A-GAL4;+* on the other. Canton-S flies were used as naive choosers. For lifespan experiments, progeny from both crosses were maintained at 18 °C throughout development. Following eclosion, females were mated for 3 days, separated by gender, and placed at 29 °C to begin lifespan measurement.

**Bacterial infection with *P. aeruginosa*.** The PA14 *plcs* strain used in this study was obtained from L. Rahme (Harvard Medical School). For each experiment, a glycerol stock was freshly streaked onto an LB/gentamicin plate. After an overnight incubation, a single colony was picked and grown in 1 ml of LB/gentamicin until this seed culture reached logarithmic phase. Subsequently, the culture was diluted in 25 ml of LB/gentamicin and grown until the desired $A_{600}$ concentration was reached. Finally, the bacterial culture was centrifuged and the pellet resuspended in LB media to obtain an $A_{600}$ reading of 100. The culture was kept on ice during infection. Needles were directly placed in the concentrated bacterial solution and then poked into the fly abdomen. After infection, flies were transferred to standard food vials and kept in the incubator at 25 °C and 60% humidity. Flies were collected 24 h post infection for behavioral experiments. Infected flies were loaded in one arm of the T connectors and control flies (not infected with *P. aeruginosa*) were loaded into the opposite arm. Twenty naive choosing flies were introduced into the central arm of the maze and the number of flies in each arm of the trap was counted at regular intervals.

**Quantification and statistical analysis.** For all preference assays, $P$ values comparing the PI among treatments was obtained using a randomization procedure and the statistical software R. Briefly, the null distribution of no difference among treatments was obtained by randomizing individual PIs obtained from groups of 20 flies among all measures (maintaining block structure where appropriate) and 100,000 $t$-statistics (or $F$-statistics for multiple comparisons). $P$ values (one-sided or two-sided as appropriate) were determined by computing the fraction of null values that were equal or more extreme to the observed $t$-statistic (or $F$-statistic). Mean preference values were plotted and weighted by the number of choosing flies in each trial, with the error bars representing the standard error of the mean (SEM). Experiment-wise error rates for experiments comparing three or more treatments were protected by presentation of treatment $P$ value from non-parametric, randomization ANOVA, which are reported in

the figure legends when appropriate. For lifespan and starvation assays, we employed survival analysis. Unless otherwise indicated, groupwise and pairwise comparisons among survivorship curves (both lifespan and starvation) were performed using the DLife computer software[63] and the statistical software R. $P$ values were obtained using the log-rank analysis (select pairwise comparisons and group comparisons or interaction studies) as noted. Interaction $P$ values were calculated using Cox regression when the survival data satisfied the assumption of proportional hazards. In other cases (as noted in the figure legends), we used ANOVA to calculate $P$ values for the interaction term for age at death. For all box plots, the box represents SEM (centered on the mean), and whiskers represent 10%/90%. For $CO_2$, TAG, and negative geotaxis measures, $P$ values were obtained by standard two-sided $t$ test after verifying normality and equality of variances. Details of the metabolomics analysis are presented above.

**Reporting summary.** Further information on research design is available in the Nature Research Reporting Summary linked to this article.

## Data availability

Metabolomics data and analyses are provided as Supplementary File 1. All additional data and analysis scripts that support the findings of this study are available from the corresponding author on request.

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

## Acknowledgements

We would like to acknowledge the members of the Pletcher laboratory for their comments on the experimental design and analysis. This research was supported by the US National Institutes of Health, National Institute on Aging (RO1AG030593 and RO1AG023166 to S.D.P.); the Glenn Medical Foundation (to S.D.P.), and the NIH Cellular and Molecular Biology and Career Training in the Biology of Aging Training Grants (T32-GM007315 and T32-AG000114 to A.S.M.).

## Author contributions

T.S.C., C.M.G., A.S.M., and S.D.P. designed the experiments. T.S.C., C.M.G., A.S.M., M.N.D., and Z.W.H. performed the experiments. C.M.G., A.S.M., Y.L., and S.D.P. analyzed the data. T.S.C., C.M.G., Y.L., A.S.M., and S.D.P. wrote the manuscript.

## Additional information

**Competing interests:** The authors declare no competing interests.

