## [Peer Review File · Nature Communications]

Reviewers' comments:

Reviewer #1 (Remarks to the Author):

The manuscript by Scott Pletcher and colleagues characterized a novel phenomenon that the exposure to dead conspecific flies induced aversive cues within the living flies, and modulated these flies' general metabolism, lifespan, and behavioral choice. These results suggesting the possible existence of death perception in fruit flies. The authors also showed that visual and olfactory cues, as well as 5HT signaling, were critical for some of these effects.

Overall, although the authors indeed observed a very interesting phenomenon, their study is largely observational and incomplete. I would expect a more thorough and in-depth study before it is published in any high profile journals.

Specifically, to call it "death perception", the authors should at least show which information from dead flies is actually "perceived". The authors only showed that visual and olfactory cues were required. Were they also sufficient to induce death perception? The authors showed that naturally dead flies, but not those killed by freezing, were sufficient to induce aversive cues. But supposedly these dead flies look similar to each other. How would the authors reconcile this with the requirement of visual cues in death perception? After the exposure to death flies, which kind of aversive cues are induced and emitted from living flies? Without answering these questions, it would be hard to really establish the concept of "death perception" in flies.

In addition, mechanistically, the involvement of 5HT signaling in death perception is rather weak. Are 5HT+ neurons necessary and/or sufficient for "death perception"? Is 5HT signaling involved in the perception of sensory cues from dead flies, or in the regulation of general sensory perception per se? Is it involved in the regulation of lifespan per se (btw the lifespan data for 5HT2A mutants are rather noisy and weak)?

Reviewer #2 (Remarks to the Author):

In this work, the investigators report that flies avoid zones with a combination of dead plus live flies, in favor of a zone with live flies only. Naïve flies even avoid flies that have been exposed to dead flies (no dead flies present). This avoidance behavior appears to be visual, as well as olfactory in nature. Remarkably, if the flies were kept in the dark, they did not avoid dead flies. Exposed flies displayed reduced climbing and lifespan. The authors also performed metabolomics and found that test flies exposed to dead flies showed changes in chemicals, some of which are associated with mood disorders in humans. Finally, they found that an antagonist of the serotonin receptor, pirenperone, which is used in humans to treat anxiety, and mutation of the serotonin receptor, mitigated the effects of exposure to dead flies. This is certainly an interesting study. In fact, it is quite provocative. I have a few suggestions for improvement.

1) The introduction of this paper begins with a mention of the psychological stress experienced by animals, including humans, due to exposure to dead conspecifics. Along these lines, the premise of this work is that flies are also traumatized by exposure to dead flies. However, if that is the case, why does the aversion to dead flies disappear if the flies have been killed by freezing or have been dead for many days? These observations undermine the concept that exposure to dead flies is by nature traumatic. This needs to be discussed.

2) Another issue that undermines the claim that dead flies are aversive is that when given a choice between live flies and dead flies only, the test flies slightly preferred the dead flies. The interpretation of this experiment is that CO₂ is aversive, and only live flies release CO₂. To test this model, the investigators should examine the behavior of mutant flies missing one or both CO₂ receptors (Gr21a and Gr63a).

3) Insufficient information is provided as to how the behavioral preference assays were conducted to ascertain preferences between zones with live vs dead (or live + dead) flies. How many live and dead flies were presented on either side of the chamber?

4 The investigators terminate their preference assays after 12 minutes. Do the animals maintain their preference for live flies only for an extended period, such as 1 or 2 hours or longer? A time course would be useful here to assess whether the investigators are measuring just an acute response to the dead flies. Do they adapt after longer periods?

5) The effect of exposure to dead flies decreased lifespan dramatically, as indicated in Figure 2h. However, in Supp Fig. 3b, why is the effect so subtle?

6) Many sensory receptors and ion channels are not specific to one sense. Even *norpA* mutations (*plc*) can affect olfaction, and *Ir76b* mutations can also impact on olfaction, in addition to taste. Therefore, it is important to be careful when interpreting the data with these and other mutations,

Reviewer #3 (Remarks to the Author):

Chakraborty et al. present interesting new findings relating to the perception of death in *Drosophila*. The work is highly creative and could be of potential interest to a broad audience. In this work, the authors report that flies can perceive dead conspecifics and this leads to alterations in physiology and lifespan. The authors conclude that their work reveals conserved links between psychological state and aging and that these findings will open up new studies in this area using the fly as a model.

On the whole, the findings are intriguing. However, I have a number of technical and conceptual concerns and suggestions to improve the paper.

Figure 1: It would appear that the term 'death perception' may be too broad. How do the authors interpret the fact that long-dead flies (or frozen flies) do not elicit a response? They are dead.

Figure 2: the data showing alterations in starvation resistance, TAG and CO₂ are interesting. However, it seems possible that this may result from alterations in bacterial levels associated with dead flies. Indeed, infected flies display a 'metabolic wasting phenotype' Dionne et al. 2006 *Curr. Bio.*

Do axenic dead flies induce this response? This seems an important control as the authors are seeking to link the changes to perception.

Isn't this a concern for Fig 2H lifespan data also?

Figure 3: The data in 3b are very interesting. How many replicates have been performed? This is perhaps the strongest data to support the idea that perception is key (although confounding effects related to bacteria could still play a role). It would be important to show replicate experiments.

In 3d, it seems that blind flies do display shortened lifespan in the presence of dead flies. The authors should be very careful how to interpret this result.

Conceptual concern: If taken on face value, in the authors' model, shouldn't blind flies be long-lived under standard lab conditions? ie if seeing dead flies shortens lifespan?

This apparent paradox may warrant discussion.

Figure 4: the data in panels A and B are potentially interesting. However, it is unclear if any of the reported changes cause the reported phenotypes. A related technical concern is that *norpA* flies display reduced lifespan upon exposure to dead flies (Fig 3d). So, again, the interpretation is complicated. The authors may wish to discuss this.

That said, if the authors wish to strengthen the conclusions, they could show that these changes are also abolished in flies in dark.

The data in Fig 4d are interesting. But, once again, the interpretation is complicated and, on face value, not consistent with the authors model. As I understand it, the authors hypothesis is that 'death perception' induces an 'anxiety state', which shortens lifespan. How do the authors reconcile this with the lifespan shortening effects of pirenperone?

Again, in standard 'lifespan assays' flies encounter dead conspecifics. Shouldn't this treatment (pirenperone) protect against this?

In any case, it is difficult to interpret the result as the treatment shortens lifespan. At the very least, the data should be interpreted with caution.

The data in 4f do not clearly support the overall claims. The lifespan of the mutant flies is shortened by exposure to dead flies. The magnitude of the effect may be reduced. But, the result appears to be significant.

A key point that should be resolved is the potential role of microbes as a potential confounding effect. I will present a hypothetical scenario to illustrate: What if 5HT2A mutants have an elevated microbial load and, as a result, reduced lifespan? What if exposure to dead flies leads to elevated microbial load and shortened lifespan?

The manuscript would be greatly improved if the authors could exclude the potential role of microbes in mediating the detrimental effects of exposure to dead flies.

Technical question: is there a way to induce shortened lifespan by just showing dead flies, ie. not placing in the same vial?

Response to Individual Referees comments:

Reviewer #1:

Overall, although the authors indeed observed a very interesting phenomenon, their study is largely observational and incomplete. I would expect a more thorough and in-depth study before it is published in any high profile journals. Specifically, to call it "death perception", the authors should at least show which information from dead flies is actually "perceived". The authors only showed that visual and olfactory cues were required. Were they also sufficient to induce death perception?

In our opinion, a full understanding of the mechanisms underlying the death cue, its perception, and its effects across the organism lie outside the scope of a single publication. The evidence provided in our manuscript suggests a complex set of cues likely involving more than sensory modality (sight and smell). While testing olfactory cues may be approachable, visual cues are not. More importantly, to definitively identify a cue, one must establish it as sufficient to mimic the effects of death exposure, such as aversiveness and reduced lifespan. However, these same phenotypes will result from manipulations that generate sick flies independent of death perception. Therefore, one must understand both neural mechanism and environmental cue to say anything definitive about either. These goals are well outside the scope of a single manuscript, and we would argue that there are few instances of sensory perception where such clear details are known in any system. We believe that our evidence strongly suggests the existence of such a perceptive ability, and that our characterization of its impact on relevant phenotypes is novel and important. Our identification of at least two biological pathways involved are important scientific advances that should be made available to the scientific community. They set the stage for us and our *Drosophila* colleagues to identify detailed mechanisms, and they provide new avenues of investigation into the psychological determinants of mammalian aging.

In the revision we have tempered the assertion of perception somewhat (i.e., first calling it a "putative perceptive experience.") although the title and final two paragraphs still refer to death perception. We would be willing to consider rephrasing this, although our data do clearly show phenotypes that require sensory modalities.

The authors showed that naturally dead flies, but not those killed by freezing, were sufficient to induce aversive cues. But supposedly these dead flies look similar to each other. How would the authors reconcile this with the requirement of visual cues in death perception?

As noted above, without a detailed understanding of the nature of the cue(s) involved, as well as the neural circuitry required to detect them, this is impossible to answer definitively. Nevertheless, we do present pictures of flies that died naturally and as a result of freezing (Supp Fig. 1), and we note their distinct appearance. We leave it up to the reader to decide how such differences may or may not be interpreted by the fly.

After the exposure to death flies, which kind of aversive cues are induced and emitted from living flies?

Olfactory cues (at least) are involved. In this revision, we provide new data that show that the CO₂ co-receptor, *Gr63a*, is required for naïve flies to detect aversive cues emitted from animals exposed

to dead conspecifics (Fig. 2a). As we noted in the original version, *Orco* is also required, suggesting that multiple odorant cues are emitted following death perception.

In addition, mechanistically, the involvement of 5HT signaling in death perception is rather weak. Are 5HT+ neurons necessary and/or sufficient for "death perception"?

We have added new data to the revision showing that acute activation of 5-HT_{2A}⁺ neurons is sufficient to induce aversiveness and chronic activation is sufficient to reduce lifespan (Fig. 5). These results are consistent with the notion that this manipulation mimics death perception. However, for reasons mentioned above and discussed in the revision, they are also open to alternative interpretation because both phenotypes might be observed following non-specific manipulations that result in sick flies. So, while insightful, we interpreted these data with caution.

Is 5HT signaling involved in the perception of sensory cues from dead flies, or in the regulation of general sensory perception per se? Is it involved in the regulation of lifespan per se (btw the lifespan data for 5HT_{2A} mutants are rather noisy and weak)?

We find no evidence that the 5-HT_{2A} receptor is required for olfactory or visual processing. On the other hand, its role in sensory modulation of aging and physiological health is emerging as a common theme across different experimental models. We have addressed this question by adding significant text in the Discussion.

Reviewer #2:

In this work, the investigators report that flies avoid zones with a combination of dead plus live flies, in favor of a zone with live flies only. Naïve flies even avoid flies that have been exposed to dead flies (no dead flies present). This avoidance behavior appears to be visual, as well as olfactory in nature. Remarkably, if the flies were kept in the dark, they did not avoid dead flies. Exposed flies displayed reduced climbing and lifespan. The authors also performed metabolomics and found that test flies exposed to dead flies showed changes in chemicals, some of which are associated with mood disorders in humans. Finally, they found that an antagonist of the serotonin receptor, pirenperone, which is used in humans to treat anxiety, and mutation of the serotonin receptor, mitigated the effects of exposure to dead flies. This is certainly an interesting study. In fact, it is quite provocative. I have a few suggestions for improvement.

1) The introduction of this paper begins with a mention of the psychological stress experienced by animals, including humans, due to exposure to dead conspecifics. Along these lines, the premise of this work is that flies are also traumatized by exposure to dead flies. However, if that is the case, why does the aversion to dead flies disappear if the flies have been killed by freezing or have been dead for many days? These observations undermine the concept that exposure to dead flies is by nature traumatic. This needs to be discussed.

We detect some confusion in certain aspects of this reviewer's understanding of our experiments and data. Specifically, we do not present evidence that naive flies avoid dead flies. Instead, we show that flies that have been exposed to naturally dead flies (*i*) emit olfactory cues that are detected as aversive cues by naive flies and (*ii*) undergo a set of physiological changes that compromise health and aging. If flies are exposed to dead flies in the dark, or if the dead flies were killed by freezing or were long dead, then aversive cues are not emitted from exposed animals and lifespan/health phenotypes are not observed. We interpret these results as the exposed flies simply failing to experience the perception of dead individuals, either because they did not detect a required cue (e.g., in the dark) or the cue(s) were not present (e.g., freezing death or long-dead animals). We worked diligently to clarify this in the revision.

*2) Another issue that undermines the claim that dead flies are aversive is that when given a choice between live flies and dead flies only, the test flies slightly preferred the dead flies. The interpretation of this experiment is that CO₂ is aversive, and only live flies release CO₂. To test this model, the investigators should examine the behavior of mutant flies missing one or both CO₂ receptors (*Gr21a* and *Gr63a*).*

Again, we do not claim (or believe) that dead flies are aversive, and the experiment to which the reviewer refers was a control manipulation to account for potential experimental artifacts. Nevertheless, we have added new data showing that *Gr63a* is required for naïve flies to detect olfactory cues emitted from flies exposed to dead conspecifics (Fig 2a). Incidentally, when given the choice between only live flies and only dead flies, *Gr63a* mutants do not exhibit a preference, which supports our original interpretation of this control experiment (data not presented).

3) Insufficient information is provided as to how the behavioral preference assays were conducted to ascertain preferences between zones with live vs dead (or live + dead) flies. How many live and dead flies were presented on either side of the chamber?

We have added this information to the Methods section. Unless otherwise noted, all behavioral choice experiments were carried out by exposing 20 live flies to 14 dead flies, which were the conditions that provided the most robust results (see Fig. 1e).

4) The investigators terminate their preference assays after 12 minutes. Do the animals maintain their preference for live flies only for an extended period, such as 1 or 2 hours or longer? A time course would be useful here to assess whether the investigators are measuring just an acute response to the dead flies. Do they adapt after longer periods?

Again, as noted above, we are not testing whether flies exhibit a preference for live flies (they do not). The experiments to which this reviewer refers test whether flies exposed to dead animals produce aversive cues in response to that experience. The data presented in the manuscript show that the response is not acute, but it is short-lived. In our experiments, we show that the aversive cues are emitted even following 72 hrs of exposure to dead animals (see Fig. 1d), suggesting that the flies don't "adapt" rapidly to the presence of dead animals. However, once dead flies are removed, the aversive characteristic of flies exposed to dead conspecifics is lost after roughly 10 minutes (Fig. 1c), suggesting that they "forget" about the experience rather quickly.

5) The effect of exposure to dead flies decreased lifespan dramatically, as indicated in Figure 2h. However, in Supp Fig. 3b, why is the effect so subtle?

As is common for lifespan phenotypes, there is considerable variation in the effects of different treatments in different genetic backgrounds. Indeed, we have many years of experience dealing with these issues with respect to lifespan and other aging phenotypes. With the exception of the *norpA* flies (also noted in the revision), all mutant alleles were extensively backcrossed to an appropriate laboratory controlled stock (either Canton-S or *w¹¹¹⁸*). These stocks served as contemporaneous background controls in each experiment. Unfortunately, it is not practical to cross all mutant strains to a single genetic background because of differences among stocks in allele markers. For example, mutants marked by the mini-white gene are not easily backcrossed into a Canton-S strain with wild-type eyes. Following mutant alleles via PCR is possible, but it introduces problems of its own, not least of which are the severe population bottlenecks that inevitably happen during the backcrossing and re-isolation of homozygous mutant flies. It is important to note that we were careful to measure the aversive and lifespan responses to dead individuals in all of the backgrounds used as controls in the manuscript. We found that all are affected, albeit some to slightly lesser extent.

*6) Many sensory receptors and ion channels are not specific to one sense. Even *norpA* mutations (*plc*) can affect olfaction, and *Ir76b* mutations can also impact on olfaction, in addition to taste. Therefore, it is important to be careful when interpreting the data with these and other mutations,*

We appreciate these concerns and have worked hard in the revision to temper our interpretation of the sensory experiments.

Reviewer #3:

Chakraborty et al. present interesting new findings relating to the perception of death in Drosophila. The work is highly creative and could be of potential interest to a broad audience. In this work, the authors report that flies can perceive dead conspecifics and this leads to alterations in physiology and lifespan. The authors conclude that their work reveals conserved links between psychological state and aging and that these findings will open up new studies in this area using the fly as a model. On the whole, the findings are intriguing. However, I have a number of technical and conceptual concerns and suggestions to improve the paper. Figure 1: It would appear that the term 'death perception' may be too broad. How do the authors interpret the fact that long-dead flies (or frozen flies) do not elicit a response? They are dead.

As noted above in our response to Reviewer #1, we really don't know. We interpret this as a result of what appears to be a complex set of cues that are required for recognizing the animals as dead conspecifics. Both olfaction and vision seem to be involved, and there are distinct differences in appearance of naturally dead vs frozen flies (see Supp Fig 1). Long-dead flies also appear different to the human eye. Without a detailed understanding of the nature of the cue(s) involved, as well as the neural circuitry required to detect them, this is impossible to answer definitively. We have added some discussion of this to the revision and prefer to leave it up to the reader to decide how such differences may or may not be interpreted by the fly.

Figure 2: the data showing alterations in starvation resistance, TAG and CO2 are interesting. However, it seems possible that this may result from alterations in bacterial levels associated with dead flies. Indeed, infected flies display a 'metabolic wasting phenotype' Dionne et al. 2006 Curr. Bio. Do axenic dead flies induce this response? This seems an important control as the authors are seeking to link the changes to perception. Isn't this a concern for Fig 2H lifespan data also?

This is an excellent suggestion by the reviewer. We report in the revision (Supp Fig 2f and respective Methods) that effects of death are unchanged when axenic flies are used for both exposed and dead animals. Moreover, we note in the manuscript that key environmental and genetic manipulations (e.g., exposure in the dark, using *norpA* strains) were capable of eliminating the effect, effectively ruling out bacteria and other common microbes as a key component of this response.

Figure 3: The data in 3b are very interesting. How many replicates have been performed? This is perhaps the strongest data to support the idea that perception is key (although confounding effects related to bacteria could still play a role). It would be important to show replicate experiments.

This experiment has been repeated several times. We present data from a second replicate as new Supp Fig. 4a.

In 3d, it seems that blind flies do display shortened lifespan in the presence of dead flies. The authors should be very careful how to interpret this result.

We note that there is a statistically significant interaction between exposure and genotype suggesting that, although *norpA* flies do respond, they do so significantly less well (see also Supp Fig.

4b). This is, indeed, a point that we have now emphasized as it suggests that *norpA* is only partly required. We should note also that *norpA* loss of function fully reversed the aversive phenotype.

Conceptual concern: If taken on face value, in the authors' model, shouldn't blind flies be long-lived under standard lab conditions? ie if seeing dead flies shortens lifespan? This apparent paradox may warrant discussion.

On the surface, yes, one might expect blind flies to be long-lived. However, this assumes that they would not suffer other deleterious pleiotropic effects, such as associate neural defects or problems navigating the environment. Avoiding ancillary causes of death is difficult, but there is some evidence provided by Fig 3b and Supp Fig. 4a, showing that (unexposed) flies aged in the dark (i.e., those that are effectively blind) do live longer than those kept under light-dark conditions. This point is mentioned in the revision.

*Figure 4: the data in panels A and B are potentially interesting. However, it is unclear if any of the reported changes cause the reported phenotypes. A related technical concern is that *norpA* flies display reduced lifespan upon exposure to dead flies (Fig 3d). So, again, the interpretation is complicated. The authors may wish to discuss this. That said, if the authors wish to strengthen the conclusions, they could show that these changes are also abolished in flies in dark.*

We are clear in the manuscript that the observed differences in the neurometabolome represent correlations, and we did not mean to infer that they are causal. Indeed, at this point we do not believe that such an experiment is likely to reveal specific metabolites that are causal, simply because we found little in the way of statistical interaction despite a good deal of statistical power (8 biological replicates for each of four treatments, N=32 total). It is likely that many small, but directed, changes in the brain are responsible for the phenotypic effects we describe. Nevertheless, we believe that these data are useful. They clearly identified a neural signature of death perception (Figure 4a), which is, in our opinion, a remarkable result. It quantifies a definitive, short-term systems-level neuronal response to dead individuals. We also now consider the partial response of *norpA* in interpreting these data. We believe that this approach has the potential to open future directions for a systems biology approach to the problem.

The data in Fig 4d are interesting. But, once again, the interpretation is complicated and, on face value, not consistent with the authors model. As I understand it, the authors hypothesis is that 'death perception' induces an 'anxiety state', which shortens lifespan. How do the authors reconcile this with the lifespan shortening effects of pirenperone? Again, in standard 'lifespan assays' flies encounter dead conspecifics. Shouldn't this treatment (pirenperone) protect against this? In any case, it is difficult to interpret the result as the treatment shortens lifespan. At the very least, the data should be interpreted with caution.

As with many epistasis experiments in aging biology, there are often pleiotropic effects of the manipulation that act in parallel to the pathway in question. This does not necessarily invalidate important inference. Pirenpirone (as well as *5-HT2A* genetic loss of function) may have broader consequences than simply interfering with death perception. The question of requirement does not associate with whether the manipulations affect lifespan on their own (in many cases, such as *Orco*

mutants, we have already published that they do!), but whether the manipulations influence the ability of death perception to affect lifespan *independent of the main effects of the genetic manipulation*. The classic identification of the transcription *daf-16/FOXO* as required for the lifespan extension observed in *daf-2* mutant *C. elegans* was successful despite strongly pleiotropic effects of loss of *daf-16*. This type of inference was clearly presented in the manuscript through the statistical interaction between genotype and death exposure, which we have provided. Whether 5-HT2A activation is sufficient to mimic death perception is a much more difficult question (see responses to reviewers #1 and #2 above), although the new data that we present in the revision are consistent with such an interpretation. We have added text in the revision to acknowledge this reviewer's concern.

The data in 4f do not clearly support the overall claims. The lifespan of the mutant flies is shortened by exposure to dead flies. The magnitude of the effect may be reduced. But, the result appears to be significant.

The effect of dead flies is significantly reduced in 5-HT2A mutants, as noted in the Figure legend, providing evidence of its involvement in this phenotype. We leave it to the reader to determine whether it can be considered "significant" with $P = 0.06$.

A key point that should be resolved is the potential role of microbes as a potential confounding effect. I will present a hypothetical scenario to illustrate: What if 5HT2A mutants have an elevated microbial load and, as a result, reduced lifespan? What if exposure to dead flies leads to elevated microbial load and shortened lifespan? The manuscript would be greatly improved if the authors could exclude the potential role of microbes in mediating the detrimental effects of exposure to dead flies.

As noted above, several lines of evidence rule out microbes as key components of this response: 1) aversive phenotypes persisted when bacteria/microbes were removed through axenic culturing, 2) exposure in the dark eliminated the effects of exposure (microbes were presumably unaffected by 48hrs of darkness), 3) other genetic manipulations (e.g., *Orco* mutation and *norpA*) were also capable of reversing the effects without any differences in microbial environment.

Technical question: is there a way to induce shortened lifespan by just showing dead flies, ie. not placing in the same vial?

Unfortunately, we have been unable to induce short lifespan in this manner.

Reviewers' comments:

Reviewer #1 (Remarks to the Author):

After carefully examining the revised manuscript and the rebuttal letter, I am still not fully convinced that the authors are indeed reporting a "death perception" phenotype. As I mentioned in the first round of review, the authors, to the minimum, should have some ideas about what signals from the dead flies are actually "perceived" by the living flies. From the present data, one can barely say that visual input is necessary as well as the olfactory cues are required, but this does not necessarily mean that flies are detecting any kind of death-specific signals via visual and/or olfactory systems,

I agree with the reviewers that the visual system is quite hard to dig further, but for the olfactory system, ample mutant and GAL4 reagents are present for a more detailed and thorough examination. To the very least, even if the authors cannot identify the exact sensory cues that convey death, they need to demonstrate that visual/olfactory input are sufficient to induce such "death perception". Just like Reviewer 3 also pointed out, by placing dead flies together with living flies in a same vial is too messy a treatment. A lot of things can happen.

Again, the difference between flies that died naturally and those killed by freezing is not clear. From Fig. S1, I guess by human eyes those flies may look different, but human perceptions do not matter here. It is really hard to guess whether the flies can see something different. If I would guess, one more plausible explanation is that, flies that died naturally, but not those killed by freezing, may release some short-range pheromones as alarm signals and trigger the phenotypes seen by the authors. This unknown pheromone only works in close proximity so that the visual input is required to attract flies. This model may simply be wrong. But the point is, the authors should consider alternative explanations more seriously before drawing a rather eye-catching new concept like "death perception in flies".

Also the physiological significance of this "death perception" phenomenon is unclear. As other reviewers mentioned, living flies encounter dead flies quite often in their life (both in lab and in nature), why should they have evolved such a mechanism?

As for the mechanistic study, the authors did add a few new dataset to their study. It is more convincing now that 5HT2A is involved (although again it lacks gain of function data). But since the major conceptual advance of this story is the identification and characterization of a novel "death perception" phenomenon, I believe that the present manuscript has not reached the standard to publish in Nature Communications, because of the reasons mentioned above.

Reviewer #2 (Remarks to the Author):

As indicated in my initial review, this work on "death perception" is provocative. All three reviewers raise the same obvious problem that if the animals are perceiving dead flies, then frozen, or long-dead flies should also be recognized as dead. If the differences are due to physical distinctions between dead flies that have been frozen, then it would have been helpful to if the authors were able to address the last comment by reviewer #3: "is there a way to induce shortened lifespan by just showing dead flies, ie. not placing in the same vial?" Without being able to resolve this issue concerning the frozen or long-dead flies, the overall interpretation of their findings is not on solid ground. While I commend the authors for their creativity, and thinking outside the box, I question whether they are actually studying "death perception" per se.

Reviewer #3 (Remarks to the Author):

Chakraborty et al. present interesting new findings relating to the perception of death in *Drosophila*. The work is highly creative and could be of potential interest to a broad audience. In this work, the authors report that flies can perceive dead conspecifics and this leads to alterations in physiology and lifespan. The authors

conclude that their work reveals conserved links between psychological state and aging and that these findings will open up new studies in this area using the fly as a model. On the whole, the findings are intriguing. However, a number of technical and conceptual concerns remain unanswered in this revised manuscript.

1) It would appear that the term 'death perception' may be too broad. Long-dead flies (or frozen flies) do not elicit a response? They are dead.

2) It seems possible that the reported effects may involve alterations in bacterial levels associated with dead flies. Indeed, infected flies display a 'metabolic wasting phenotype' Dionne et al. 2006 Curr. Bio. Also, the gut microbiota has been shown to impact lifespan in flies and vertebrates.

Do axenic dead flies induce metabolic wasting (changes in TAG/starvation resistance)? Do axenic dead flies shorten lifespan?

These questions from the original review appear to remain unanswered.

This seems an important control as the authors are seeking to link the changes to perception.

Responses to Reviewers

Reviewer #1

“After carefully examining the revised manuscript and the rebuttal letter, I am still not fully convinced that the authors are indeed reporting a "death perception" phenotype. As I mentioned in the first round of review, the authors, to the minimum, should have some ideas about what signals from the dead flies are actually "perceived" by the living flies. From the present data, one can barely say that visual input is necessary as well as the olfactory cues are required, but this does not necessarily mean that flies are detecting any kind of death-specific signals via visual and/or olfactory systems,

I agree with the reviewers that the visual system is quite hard to dig further, but for the olfactory system, ample mutant and GAL4 reagents are present for a more detailed and thorough examination. To the very least, even if the authors cannot identify the exact sensory cues that convey death, they need to demonstrate that visual/olfactory input are sufficient to induce such "death perception".

Again, the difference between flies that died naturally and those killed by freezing is not clear. From Fig. S1, I guess by human eyes those flies may look different, but human perceptions do not matter here. It is really hard to guess whether the flies can see something different. If I would guess, one more plausible explanation is that, flies that died naturally, but not those killed by freezing, may release some short-range pheromones as alarm signals and trigger the phenotypes seen by the authors. This unknown pheromone only works in close proximity so that the visual input is required to attract flies. This model may simply be wrong. But the point is, the authors should consider alternative explanations more seriously before drawing a rather eye-catching new concept like "death perception in flies".

We designed and manufactured a set of new exposure chambers that allowed us to test sufficiency of sight, smell, and taste as well as to determine whether specific sensory cues distinguish between live and dead flies. Remarkably, we show that short-term (i.e., 48hr) exposure to the sight of flies that had died naturally (through starvation, for example) is sufficient to induce the aversive phenotype. The sight of flies that had been killed by immersion in liquid nitrogen had no effect, thus replicating our earlier results indicating that flies have the ability to visually distinguish differences in these two types of corpses and that flies killed by freezing lack one or more key visual characteristics that are present in flies that died naturally. The ability of flies to distinguish differences by sight may also explain why live *D. melanogaster* do not convey aversive signals when exposed to dead *D. virilis*, as these flies are darker and significantly larger than *D. melanogaster* themselves. Neither smell of naturally dead flies nor taste of extracts from naturally dead animals is sufficient to induce phenotypes associated with death perception. We also show that olfactory cues from naturally dead animals are incapable of gating otherwise insufficient visual cues from animals that had died by freezing.

New data supporting these inferences are now presented as: Figs 3e, 3f, 3g, 3h, Supplementary Figs 4a, 4b, 6a, 6b, 6c, 6d, 7b

We summarize these experimental results, and our responses to these critiques, in the manuscript as follows:

*“Together, these data indicate that there is an essential perceptual component associated with the physiological and health effects of exposure to dead conspecifics. Sight of naturally dead conspecifics is both necessary and sufficient to induce physiological effects, suggesting a model in which visual cues serve as the primary way in which *D. melanogaster* distinguish their dead and adding to a growing literature indicating that flies are capable of extracting and responding to different ecologically-relevant visual cues in their environment, such as parasites and competitors (Schneider et al, 2018; Kacsoh et al., 2013). While gustatory cues are apparently not involved in the effects that we observe, the role of olfaction is less clear. Smell-deficient flies respond less strongly to dead individuals, but odors from dead flies are not sufficient to induce measurable changes in aversiveness. Interestingly, the aversive cues emitted by healthy flies following exposure to dead individuals have a significant olfactory component: when *Orco*² mutants were used as naïve choosers in the T-maze assay (e.g., Fig. 1), they assorted randomly between the two arms (Supplementary Fig. 8a). Similar results were observed*

when naïve choosers carried a loss of function mutation for Gr63a, an essential component of the Drosophila CO₂ receptor (Supplementary Fig. 8). We therefore currently favor a model in which olfaction mediates a social cue among exposed animals that magnifies the effects of visual death perception.”

Schneider, J., Murali, N., Taylor, G. W. & Levine, J. D. Can Drosophila melanogaster tell who's who? PLoS One 13, e0205043, doi:10.1371/journal.pone.0205043 (2018).

Kacsoh, B. Z., Lynch, Z. R., Mortimer, N. T. & Schlenke, T. A. Fruit flies medicate offspring after seeing parasites. Science 339, 947-950, doi:10.1126/science.1229625 (2013).

Also the physiological significance of this "death perception" phenomenon is unclear. As other reviewers mentioned, living flies encounter dead flies quite often in their life (both in lab and in nature), why should they have evolved such a mechanism?

This is not a question that we can answer experimentally, although we speculate that exposure to dead conspecifics is a form of psychological stress that is induced by indicators of danger and that this results in the physiological effects we observe. The physiological effects seen with death perception is consistent with the effects of other types of 100% sensory input, including perception of food and mates (e.g., Libert et al, 2007; Gendron et al, 2014; Harvanek et al., 2017).

Libert S, Zwiener J, Chu X, Vanvoorhies W, Roman G, Pletcher SD: Regulation of Drosophila life span by olfaction and food-derived odors. *Science* 2007, 315(5815):1133-1137.

Gendron CM, Kuo TH, Harvanek ZM, Chung BY, Yew JY, Dierick HA, Pletcher SD: Drosophila life span and physiology are modulated by sexual perception and reward. *Science* 2014, 343(6170):544-548.

Harvanek ZM, Lyu Y, Gendron CM, Johnson JC, Kondo S, Promislow DEL, Pletcher SD: Perceptive costs of reproduction drive ageing and physiology in male Drosophila. *Nat Ecol Evol* 2017, 1(6):152.

As for the mechanistic study, the authors did add a few new dataset to their study. It is more convincing now that 5HT2A is involved (although again it lacks gain of function data).

In the previous revision we specifically added new data in response to this reviewer's request for sufficiency data. We showed that activation of 5HT2A neurons is sufficient to recapitulate the lifespan and aversion phenotypes associated with exposure to dead flies (new Fig. 6).

Reviewer #2

As indicated in my initial review, this work on "death perception" is provocative. All three reviewers raise the same obvious problem that if the animals are perceiving dead flies, then frozen, or long-dead flies should also be recognized as dead. If the differences are due to physical distinctions between dead flies that have been frozen, then it would have been helpful if the authors were able to address the last comment by reviewer #3: "is there a way to induce shortened lifespan by just showing dead flies, ie. not placing in the same vial?" Without being able to resolve this issue concerning the frozen or long-dead flies, the overall interpretation of their findings is not on solid ground. While I commend the authors for their creativity, and thinking outside the box, I question whether they are actually studying "death perception" per se.

Please see our response to Reviewer #1 about visual cues that distinguish dead from alive flies and flies that died naturally from those frozen.

Regarding the request to show a change in lifespan by showing dead flies: Unfortunately, this is technically not possible at the moment. Our new exposure chambers are excellent for single, short-term exposure events but they do not allow for replacement of dead flies and reliable transfer of live flies three times a week over the three months that constitute the full lifespan of a fly. We have yet been unable to design a chamber that effectively satisfies both of these criteria.

Reviewer #3

Chakraborty et al. present interesting new findings relating to the perception of death in Drosophila. The work is highly creative and could be of potential interest to a broad audience. In this work, the authors report that flies can perceive dead conspecifics and this leads to alterations in physiology and lifespan. The authors conclude that their work reveals conserved links between psychological state and aging and that these findings will open up new studies in this area using the fly as a model. On the whole, the findings are intriguing. However, a number of technical and conceptual concerns remain unanswered in this revised manuscript.

1) It would appear that the term 'death perception' may be too broad. Long-dead flies (or frozen flies) do not elicit a response? They are dead.

Please see our response to Reviewer #1. Yes, they are dead, but it is now apparent that they and frozen flies lack a key visual cue that induces the observed phenotypes.

2) It seems possible that the reported effects may involve alterations in bacterial levels associated with dead flies. Indeed, infected flies display a 'metabolic wasting phenotype' Dionne et al. 2006 Curr. Bio. Also, the gut microbiota has been shown to impact lifespan in flies and vertebrates. Do axenic dead flies induce metabolic wasting (changes in TAG/starvation resistance)? Do axenic dead flies shorten lifespan?

In the previous revision we specifically added new data in response to this reviewer's request investigation of gut biota. These new data (Supp Fig 2F) use axenic flies to explicitly rule out infection and gut microbiota. This is consistent without new data showing that sight of dead flies is sufficient to induce aversive phenotypes. Microbiota changes are not required for the phenotypes that we report.

Reviewers' comments:

Reviewer #1 (Remarks to the Author):

The authors did provide a new set of data showing that simply by seeing naturally dead flies (but not those killed by freezing) through a window can evoke a series of behavioral responses. This is a very strong argument that death perception via the visual system does exist.

Honestly I am still a bit skeptical about this phenomenon itself (do flies really have such good vision?) But I would support the publication of this story and hope to see more follow-up discussions and publications in the field. It is nevertheless a very provoking idea and finding and it worth being seen and discussed by the entire field.

Reviewer #2 (Remarks to the Author):

It makes sense that flies would find dead flies aversive, since it would promote survival to avoid an environment in which their conspecifics cannot survive. In principle, this is quite similar to avoiding other potentially noxious environments laden with dangerous chemicals or inhospitable temperature. I am now convinced that the authors have shown that flies find dead conspecifics aversive, and established the key sensory cues relevant to this behavior. However, it is ridiculous and unjustified to place their study in the context of "psychological trauma," "mental health issues including depression and post-traumatic stress disorder," "emotion," "post-traumatic stress disorder and depression," and "psychological experience." These statements are not limited to the Abstract, Introduction and Discussion. Even in the Results section, one set of experiments is introduced in the context of, "the severity of behavioral and emotional reactions to dead animals correlate positively with the extent that one identifies with the deceased." Another section in the Results is introduced by, "we next sought to investigate whether this experience elicited broader physiological and health effects."

Overall, the writing in this manuscript is hyperbolic and sensationalistic, and is unscientific. The authors cannot possibly presume to know the physiological state of flies. Would the authors be prepared to state that avoidance of other aversive stimuli such as noxious odorants, tastants and temperatures induce psychological trauma? In each of these cases, including the recognition of dead conspecifics, it is most likely that these reactions are simply avoidance responses to promote survival. To set this work up, and then sell the results as a model for psychological trauma will likely bring press coverage. But, it will also promote appropriate laughter and derision directed at this work, the journal and the field of *Drosophila* neurogenetics in general. Merely toning down the hyperbole is insufficient, Unless all such statements referring to mental and psychological state are removed from this manuscript, I oppose publication of this work.

Reviewer #3 (Remarks to the Author):

Taken on face value, the data in Supp Fig 2F indicate that fly associated microbes are not required to elicit an aversive response. This is interesting. However, it remains possible that altered bacteria levels play a role 'impaired health and longevity', no?

Am I missing something? If conventionally raised dead flies are placed in vials with other flies, isn't it possible/likely that this will alter the microbiota of all flies in a vial? and, that this could impact 'health and longevity'?

Perhaps the aversion and lifespan-shortening effects are not mechanistically linked. Isn't this possible?

Do axenic dead flies induce metabolic wasting (changes in TAG/starvation resistance)? Do axenic dead flies shorten lifespan?

It seems to me if this data is not included, then at the very least this should be noted (in the discussion?) as a very plausible explanation for the observed effects on lifespan. My understanding is that the lifespan shortening effects result from chronic exposure to dead flies.

Responses to Reviewers

Reviewer #1

“The authors did provide a new set of data showing that simply by seeing naturally dead flies (but not those killed by freezing) through a window can evoke a series of behavioral responses. This is a very strong argument that death perception via the visual system does exist...Honestly I am still a bit skeptical about this phenomenon itself (do flies really have such good vision?) But I would support the publication of this story and hope to see more follow-up discussions and publications in the field. It is nevertheless a very provoking idea and finding and it worth being seen and discussed by the entire field.”

We agree that the sight-only experiment was a surprising result. While we knew that sight was required, we had speculated that it would not be sufficient, assuming that flies somehow integrated several different cues. We also agree that the phenomenon remains intriguing, and it seems likely that when the detailed neural mechanisms are unraveled, there will be complexities that we either missed or misinterpreted. Nevertheless, we have been convinced by over five years' worth of data, and what we believe is a rigorous and logical investigation of the phenotype, that the response is real. As noted in our responses to the editor and to reviewer #2, we have changed the tone of manuscript to emphasize the surprising nature of our results, the limitations in the inferences that we were able to provide, and the likelihood that future experiments will clarify the provocative nature of our interpretation. This is particularly apparent in the new Discussion paragraph copied here for reference.

*“Sensory perception has emerged as a potent modulator of energy homeostasis, tissue physiology, and aging across taxa through neuronal circuits that emanate from sensory tissues and that interface with deeper regions of the central nervous system. Exposure of *Drosophila* and *C. elegans* to food-based odorants limits the beneficial effects of dietary restriction (12,24), while perception of the opposite sex modulates lifespan through neural circuits that utilize conserved neuropeptides to establish motivation and reward (11,25,26). Loss of specific olfactory and gustatory neurons modulates lifespan and physiology and influences measures of healthy aging, including sleep and daily activity patterns (14,27-29). Our data are consistent with a new perceptive influence on healthy aging in *Drosophila*, which apparently invokes the ability to distinguish the quality and status of conspecifics in the environment. This seems remarkable to us despite reports showing sufficient visual acuity in flies to visually distinguish different ecologically-relevant cues in their environment, such as parasites and competitors (30,31,32). Nevertheless, additional experiments designed to identify the ‘death’ cue(s) and the molecular mechanisms required for effects associated with death exposure will be critical to unraveling the nature of the effects we observe.”*

Reviewer #2

It makes sense that flies would find dead flies aversive, since it would promote survival to avoid an environment in which their conspecifics cannot survive. In principle, this is quite similar to avoiding other potentially noxious environments laden with dangerous chemicals or inhospitable temperature. I am now convinced that the authors have shown that flies find dead conspecifics aversive, and established the key sensory cues relevant to this behavior. However, it is ridiculous and unjustified to place their study in the context of “psychological trauma,” “mental health issues including depression and post-traumatic stress disorder,” “emotion,” “post-traumatic stress disorder and depression,” and “psychological experience.” These statements are not limited to the Abstract, Introduction and Discussion. Even in the Results section, one set of experiments is introduced in the context of, “the severity of behavioral and emotional reactions to dead animals correlate positively with the extent that one identifies with the deceased.” Another section in the Results is introduced by, “we next sought to investigate whether this experience elicited broader physiological and health effects.”

Overall, the writing in this manuscript is hyperbolic and sensationalistic, and is unscientific. The authors cannot possibly presume to know the physiological state of flies. Would the authors be prepared to state that avoidance of other aversive

stimuli such as noxious odorants, tastants and temperatures induce psychological trauma? In each of these cases, including the recognition of dead conspecifics, it is most likely that these reactions are simply avoidance responses to promote survival. To set this work up, and then sell the results as a model for psychological trauma will likely bring press coverage. But, it will also promote appropriate laughter and derision directed at this work, the journal and the field of Drosophila neurogenetics in general. Merely toning down the hyperbole is insufficient, Unless all such statements referring to mental and psychological state are removed from this manuscript, I oppose publication of this work.

This reviewer misinterprets the basic observation of our manuscript. We do not describe a situation in which dead conspecifics are aversive; it is live flies that see dead conspecifics that are aversive and that are short-lived in a 5-HT2A dependent manner. Nevertheless, in response to this reviewer's extended comments, yes, our lab and others have shown that exposure to various noxious or attractive sensory cues can induce similar effects and that these are also influenced by molecules and signaling pathways in *Drosophila* that behavioral neurogeneticists have linked to neural states such as motivation, reward, fear and depression to name a few (only a few of the most recent cited below; see refs ¹⁻¹³).

Our hypothesis that these effects reflect the influence of a conserved central state is, therefore, reasonable and scientific; it is testable; and it is based on published studies that implicate conserved pathways that are emerging as modulators of these effects.

Nevertheless, in the revised manuscript we have removed all references to this interpretation from the Introduction and Results sections. We have also added new text to the Discussion that places our results in the broader context of sensory/neural control of aging, and we now confine our provocative interpretations to a sentence in the Abstract and a portion of the last paragraph in the Discussion, where they are clearly denoted as our speculation/hypothesis.

Reviewer #3

Taken on face value, the data in Supp Fig 2F indicate that fly associated microbes are not required to elicit an aversive response. This is interesting. However, it remains possible that altered bacteria levels play a role 'impaired health and longevity', no?

Am I missing something? If conventionally raised dead flies are placed in vials with other flies, isn't it possible/likely that this will alter the microbiota of all flies in a vial? and, that this could impact 'health and longevity'?

Perhaps the aversion and lifespan-shortening effects are not mechanistically linked. Isn't this possible?

Do axenic dead flies induce metabolic wasting (changes in TAG/starvation resistance)? Do axenic dead flies shorten lifespan?

It seems to me if this data is not included, then at the very least this should be noted (in the discussion?) as a very plausible explanation for the observed effects on lifespan. My understanding is that the lifespan shortening effects result from chronic exposure to dead flies.

We appreciate the concerns of this reviewer with respect to potential effects of the microbiota. As she/he notes, the data we present clearly rule it out as a significant causal factor in the aversiveness phenotype. It is possible (maybe even likely) that the aversiveness and lifespan effects have different mechanisms, at least in part. We are unable to directly test this using axenic flies for lifespan assays because of the sheer numbers involved; we would need to collect and maintain many tens of thousands of axenic animals for long periods of time (we need to generate thousands of dead flies on a weekly basis). It is simply not possible to do this, much less guarantee that they remain sterile for the entirety

of the experiment. However, we have no intention of avoiding this topic and now devote a full paragraph in the Results section, which is copied below and which outlines our arguments for the readers' consideration.

“We note that several lines of evidence suggest that the effects of exposure to dead animals are not caused by infection, bacterial proliferation, or changes in the gut biota in either dead or exposed animals. Similar levels of aversion were induced when axenic flies were used as both exposed and dead flies, establishing that these factors do not play a significant role in the aversiveness phenotype (Supplementary Fig. 8). It remains possible that the effects of death exposure on aversiveness and lifespan are, at least in part, mechanistically distinct, and we were unable to formally test the latter because lifespan exposure experiments comprised entirely of axenic animals is technically unfeasible. Nevertheless, it seems unlikely that these factors are of major importance because very different environmental and genetic manipulations, including: aging in the dark, genetic blindness, and genetic loss of olfaction abrogate or significantly diminish the exposure effects on lifespan. These manipulations have no known effects on microbiota and almost surely would not be expected to affect it in the same way. “

Later in the manuscript we show that loss of *5-HT2A* also abrogates the lifespan effect. Again it is unlikely this manipulation influences the microbiota in a similar way as the aforementioned manipulations.

References

- 1 Gibson, W. T. *et al.* Behavioral responses to a repetitive visual threat stimulus express a persistent state of defensive arousal in *Drosophila*. *Curr Biol* **25**, 1401-1415, doi:10.1016/j.cub.2015.03.058 (2015).
- 2 Herteleer, L. *et al.* Mood stabilizing drugs regulate transcription of immune, neuronal and metabolic pathway genes in *Drosophila*. *Psychopharmacology* **233**, 1751-1762, doi:10.1007/s00213-016-4223-z (2016).
- 3 Iliadi, K. G. The genetic basis of emotional behavior: has the time come for a *Drosophila* model? *Journal of neurogenetics* **23**, 136-146, doi:10.1080/01677060802471650 (2009).
- 4 Jiang, M. D., Zheng, Y., Wang, J. L. & Wang, Y. F. Drug induces depression-like phenotypes and alters gene expression profiles in *Drosophila*. *Brain research bulletin* **132**, 222-231, doi:10.1016/j.brainresbull.2017.06.009 (2017).
- 5 O'Kane, C. J. *Drosophila* as a model organism for the study of neuropsychiatric disorders. *Current topics in behavioral neurosciences* **7**, 37-60, doi:10.1007/7854_2010_110 (2011).
- 6 Ries, A. S., Hermanns, T., Poeck, B. & Strauss, R. Serotonin modulates a depression-like state in *Drosophila* responsive to lithium treatment. *Nat Commun* **8**, 15738, doi:10.1038/ncomms15738 (2017).
- 7 Yang, Z., Bertolucci, F., Wolf, R. & Heisenberg, M. Flies cope with uncontrollable stress by learned helplessness. *Curr Biol* **23**, 799-803, doi:10.1016/j.cub.2013.03.054 (2013).
- 8 Araujo, S. M. *et al.* Chronic unpredictable mild stress-induced depressive-like behavior and dysregulation of brain levels of biogenic amines in *Drosophila melanogaster*. *Behav Brain Res* **351**, 104-113, doi:10.1016/j.bbr.2018.05.016 (2018).
- 9 Mohammad, F. *et al.* Ancient Anxiety Pathways Influence *Drosophila* Defense Behaviors. *Curr Biol* **26**, 981-986, doi:10.1016/j.cub.2016.02.031 (2016).
- 10 Musso, P. Y., Lampin-Saint-Amaux, A., Tchenio, P. & Preat, T. Ingestion of artificial sweeteners leads to caloric frustration memory in *Drosophila*. *Nat Commun* **8**, 1803, doi:10.1038/s41467-017-01989-0 (2017).
- 11 Anderson, D. J. & Adolphs, R. A framework for studying emotions across species. *Cell* **157**, 187-200, doi:10.1016/j.cell.2014.03.003 (2014).
- 12 Anderson, D. J. Circuit modules linking internal states and social behaviour in flies and mice. *Nat Rev Neurosci* **17**, 692-704, doi:10.1038/nrn.2016.125 (2016).
- 13 Kasture, A. S., Hummel, T., Sucic, S. & Freissmuth, M. Big Lessons from Tiny Flies: *Drosophila melanogaster* as a Model to Explore Dysfunction of Dopaminergic and Serotonergic Neurotransmitter Systems. *Int J Mol Sci* **19**, doi:10.3390/ijms19061788 (2018).